# Global redox proteome and phosphoproteome analysis reveals redox switch in Akt

Zhiduan Su[1,2,13], James G. Burchfield [1,2,13], Pengyi Yang [1,3,13], Sean J. Humphrey [1,2], Guang Yang[1,2], Deanne Francis[1,2], Sabina Yasmin[4], Sung-Young Shin[5,6], Dougall M. Norris[1,2], Alison L. Kearney[1,2], Miro A. Astore[4], Jonathan Scavuzzo[1,2], Kelsey H. Fisher-Wellman[7,8], Qiao-Ping Wang[1,2,9], Benjamin L. Parker[1,2], G. Gregory Neely[1,2,9], Fatemeh Vafaee [1,3], Joyce Chiu[10,11], Reichelle Yeo[10,11], Philip J. Hogg [10,11], Daniel J. Fazakerley [1,2], Lan K. Nguyen[5,6], Serdar Kuyucak[4] & David E. James[1,2,12]*

Protein oxidation sits at the intersection of multiple signalling pathways, yet the magnitude and extent of crosstalk between oxidation and other post-translational modifications remains unclear. Here, we delineate global changes in adipocyte signalling networks following acute oxidative stress and reveal considerable crosstalk between cysteine oxidation and phosphorylation-based signalling. Oxidation of key regulatory kinases, including Akt, mTOR and AMPK influences the fidelity rather than their absolute activation state, highlighting an unappreciated interplay between these modifications. Mechanistic analysis of the redox regulation of Akt identified two cysteine residues in the pleckstrin homology domain (C60 and C77) to be reversibly oxidized. Oxidation at these sites affected Akt recruitment to the plasma membrane by stabilizing the $PIP_3$ binding pocket. Our data provide insights into the interplay between oxidative stress-derived redox signalling and protein phosphorylation networks and serve as a resource for understanding the contribution of cellular oxidation to a range of diseases.

[1] Charles Perkins Centre, The University of Sydney, Sydney, NSW 2006, Australia. [2] School of Life and Environmental Sciences, The University of Sydney, Sydney, NSW 2006, Australia. [3] School of Mathematics and Statistics, The University of Sydney, Sydney, NSW 2006, Australia. [4] School of Physics, The University of Sydney, Sydney, NSW 2006, Australia. [5] Department of Biochemistry and Molecular Biology, School of Biomedical Sciences, Monash University, Clayton, VIC 3800, Australia. [6] Biomedicine Discovery Institute, Monash University, Clayton, VIC 3800, Australia. [7] Brody School of Medicine, Physiology Department, East Carolina University, Greenville, NC, USA. [8] East Carolina Diabetes and Obesity Institute, East Carolina University, Greenville, NC, USA. [9] The Dr. John and Anne Chong Laboratory for Functional Genomics, Charles Perkins Centre and School of Life & Environmental Sciences, The University of Sydney, Sydney, NSW 2006, Australia. [10] The Centenary Institute, Newtown, NSW 2042, Australia. [11] National Health and Medical Research Council Clinical Trials Centre, The University of Sydney, Sydney, NSW 2006, Australia. [12] Sydney Medical School, The University of Sydney, Sydney, NSW 2006, Australia. [13] These authors contributed equally: Zhiduan Su, James G. Burchfield, Pengyi Yang. *email: david.james@sydney.edu.au

Reactive oxygen species (ROS) modulate cell signalling and protein function via reversible protein oxidation[1–4]. Indeed, the activity of kinases and phosphatases can be altered by redox-mediated post-translational modifications (PTMs), providing a link between ROS, phosphorylation-based signalling (phospho-signalling)[1,5–8] and physiological processes including cell growth and proliferation[4]. However, excessive ROS, or oxidative stress, is a causative factor in several diseases including diabetes, atherosclerosis, and cancer[3,9–12]. This is considered to largely result from inappropriate crosstalk between ROS and phospho-signalling[13,14]. Despite this, there remains a paucity of systematic studies dissecting the impact of ROS on cell signalling networks.

Reactive thiol groups on cysteine residues switch between reduced and oxidized states in response to redox fluctuations[15–18]. ROS-dependent cysteine modifications are either reversible (e.g. disulfide bonds, S-acylation, S-glutathionylation, S-nitrosylation, S-sulfenylation, S-sulfhydration and sulfinic acid) or irreversible (such as sulfonic acid)[19]. Reversible modifications can alter protein structure and function[2,20–22], with functional roles characterised in several studies[23,24]. However, modification of specific cysteines can have diverse outcomes depending on the target and type of modification[25], and the impact of most ROS-induced protein modifications on cell signalling remains largely unexplored.

We have developed a simple model for studying widespread cellular oxidation based on inhibition of the two major thiol-based antioxidant systems (glutathione and thioredoxin)[26]. This model allows ROS effects to be studied without the complex phenotypes that may occur in disease-specific models[27]. This model also has advantages over treatments like paraquat[28], since ROS are produced from endogenous locations.

Here we combine global analyses of the phosphoproteome, cysteine redox proteome, and total proteome of adipocytes to obtain a comprehensive view of global signalling network rewiring during oxidative stress. Our analysis reveals complex remodelling of the phosphoproteome in response to widespread oxidative stress-induced cysteine oxidation. Among the changes, we identify oxidation of critical cysteine residues in the Ser/Thr kinase Akt that are essential for its plasma membrane (PM)-recruitment and activation.

## Results

### The adipocyte redox proteome. We measured the proteome and redox proteome of adipocytes treated with either carmustine (BCNU) alone or in combination with auranofin (AF) for 2 or 24 h (Fig. 1a, Supplementary Fig. 1a). BCNU/AF impaired adipocyte cellular antioxidant defences, indicated by increased peroxiredoxin 2 and peroxiredoxin 3 dimers (Fig. 1b)[26].

We measured 13,451 reversibly oxidized cysteines (median coefficient of variation (CV) of 12.2% (Fig. 1c)), with 8,991 detected in at least two replicates (Supplementary Data 1), and treatment groups clustered distinctly by principal component analysis (PCA, Fig. 1d). Of these, only ~12% (1550) were previously reported (annotated in UniProt Knowledgebase; Supplementary Data 2). The efficacy of glutathione reducatase (Gsr) inhibition by BCNU was apparent due to decreased oxidation of C54 and C59 on Gsr (isoform 2) at all time points[29,30] (Supplementary Data 3). Similarly, C90 and C93 on mitochondrial thioredoxin (Txn2), C48 on mitochondrial peroxiredoxin-5 (Prdx5) and C32 and C35 on thioredoxin (Txn) were oxidized under BCNU/AF treatment, consistent with AF-mediated inhibition of thioredoxin reductase[31] (Supplementary Data 3).

7451 proteins were quantified (Supplementary Data 4), of which 4528 and 4204 were identified in all biological replicates of BCNU and BCNU/AF treatments, respectively (median CV 6.2%, Supplementary Fig. 1b, c). BCNU/AF treatment (24 h) resulted in

increased expression of 174 proteins and decreased expression of 1161 proteins (absolute fold change > 1.5, adjusted $p < 0.05$ by moderated $t$-test), whereas 2 h treatment had minimal effect on the proteome (Supplementary Fig. 1d and Supplementary Data 5). As such, redox measurements following treatment for 24 h were normalized to protein abundance.

Treatment with BCNU or BCNU/AF for 2 h increased global cysteine oxidation ($p < 0.0001$ by moderated t-test, Fig. 1e). Cysteine oxidation decreased in the 24 h BCNU treated cells ($p < 0.0001$ by moderated t-test, Fig. 1e), suggesting a compensatory antioxidant response[26]. In contrast, cysteine oxidation increased with 24 h BCNU/AF from 1,848 to 8,372 differentially oxidized peptides ($p < 0.0001$ by moderated $t$-test from limma R package, Fig. 1e and Supplementary Data 3). These data suggest worsening oxidative stress in response to inhibition of both major cellular antioxidant defence mechanisms for 24 h.

Direction analysis[32] identified 330 oxidized peptides from 291 proteins that were oxidized in both 2 and 24 h BCNU/AF treatments, that likely represent cysteines most sensitive to oxidative stress ($p < 0.01$ by direction analysis[32], Fig. 1f, Supplementary Data 3). 60 of these were mitochondrial proteins which was expected due to the higher pH in mitochondria, favouring the reactivity of cysteine thiolates[24,33], and because mitochondria are a primary source of ROS[34]. Gene ontology (GO) enrichment analysis of the 291 oxidized proteins revealed significant enrichment in ROS metabolic processes, translation, proteolysis, protein phosphorylation and signalling (Fig. 1g).

### The oxidative stress-regulated phosphoproteome. We next quantified global changes in the adipocyte phosphoproteome using the EasyPhos workflow[35,36]. Acute oxidative-stress treatments (1 and 2 h) were employed to avoid large changes in protein abundance. Cells were treated with BCNU/AF with and without insulin to activate growth-factor signalling networks of relevance to adipocytes. We identified more than 23,000 unique phosphorylation events in 6,079 proteins (Fig. 2a, Supplementary Data 6), with highly reproducible phosphopeptide quantification across conditions (Supplementary Fig. 2a, b). Since BCNU alone had limited influence on key signalling pathways and peptide oxidation (Supplementary Fig. 2c, Supplementary Data 3), we focused on adipocytes incubated with BCNU/AF. PCA revealed clear separation of both insulin stimulation in the first principal component (PC1), and a time-dependent separation of oxidative stress in the second (PC2, Fig. 2b). Compared to non-insulin stimulated cells, we observed a reduction in sample separation in the second component (PC2) upon insulin stimulation, indicating that the impact of oxidative stress on the phosphoproteome may be counteracted by insulin stimulation (Fig. 2b).

In the absence of insulin, BCNU/AF treatment for 1 h resulted in up-regulation of 214 and down-regulation of 799 phosphosites and 2 h treatment resulted in up-regulation of 509 and down-regulation of 1586 sites (absolute fold change > 1.5, adjusted $p < 0.05$ by moderated $t$-test, Fig. 2c, Supplementary Fig. 2d, e, Supplementary Data 7). Interestingly, oxidative stress caused greater dephosphorylation than phosphorylation and this was most pronounced in non-insulin stimulated cells (Fig. 2c). As observed by PCA (Fig. 2b), insulin attenuated oxidative-stress induced changes in global phosphorylation (Fig. 2c, Supplementary Fig. 2f, Supplementary Data 7). This was particularly apparent with phosphorylation sites that were down-regulated by oxidative-stress, consistent with a degree of reciprocity between these processes. Extended BCNU/AF treatment (2 h) increased the number of oxidative-stress regulated phosphosites markedly (Fig. 2c, d, Supplementary Data 7), indicating that prolonged stress doesn't simply increase the magnitude of change

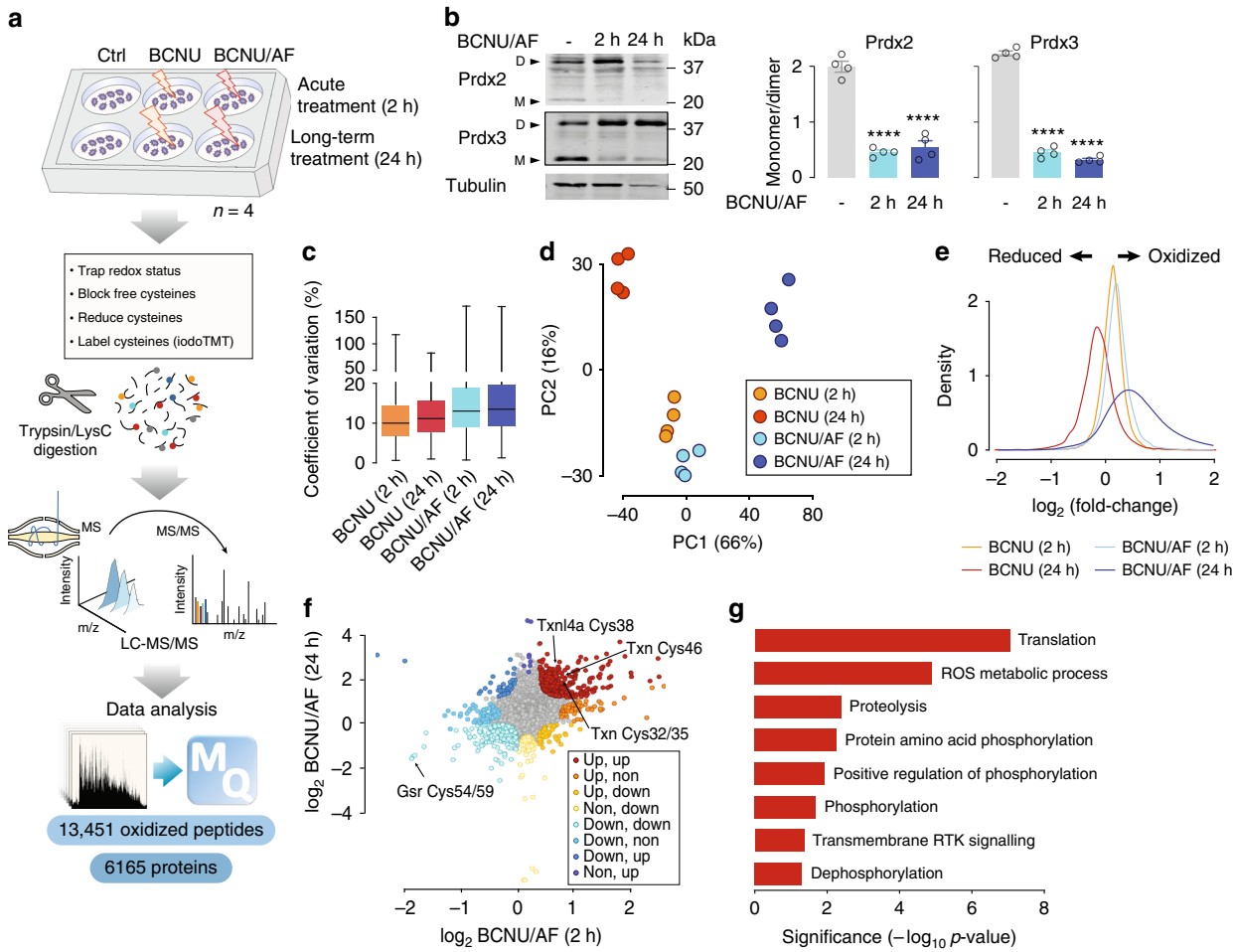

**Fig. 1** Redox proteomic analysis of adipocytes. **a** Experimental design of the redox proteomic analysis of adipocytes treated with BCNU or BCNU/AF. **b** Immunoblotting of Prdx2/3 dimerisation. Prdx2/3 dimer/monomer ratio was measured. Data were shown as mean ± SEM from $n = 4$, ****$p < 0.0001$ by one-way ANOVA. **c** Coefficient of variation of the quantified redox proteome. Boxes capture lower quartile and upper quartile with median displayed as a horizontal line in the middle; whiskers are min and max. **d** Principal component analysis of the redox proteome. **e** Relative intensities of the quantified redox proteome. **f** Direction-based integrative analysis of redox proteome between 2 h and 24 h treatment of BCNU/AF (see "Methods"). Significantly oxidised peptides were coloured ($p < 0.01$, by modified Pearson's Correlation Test[32]). **g** Direction pathway analysis of significantly oxidised proteins under both the 2 h and 24 h treatment of BCNU/AF (corresponding to the red dots in Fig. 1f) using gene ontology.

in protein phosphorylation but with increased time of exposure to oxidative stress there is a much more vast impact on the phosphoproteome than observed after acute exposure. Gene ontology enrichment analysis of proteins regulated by 2 h BCNU/AF treatment revealed the over-representation of several signalling pathways including insulin, mitogen-activated protein kinase (MAPK), receptor tyrosine-protein kinase ErbB (ErbB), mammalian target of rapamycin (mTOR), 5′ AMP-activated protein kinase (AMPK) and HIF-1 signalling (Fig. 2e). Collectively, these data indicate multiple phospho-signalling networks contain oxidant-sensitive proteins, suggesting that our dataset may allow us to identify key protein oxidation events that fine tune signalling through these major signalling nodes.

**Crosstalk of oxidative stress and phospho-signalling network.** Integrative analysis of our multi-omics datasets revealed protein oxidation to be negatively correlated with protein abundance following 24 h BCNU/AF treatment ($r = -0.37$, $p < 2.2e-16$ by Pearson's correlation, Supplementary Fig. 3a). This was more pronounced when only considering annotated kinases and substrates ($r = -0.42$, $p < 2.2e-16$ by Pearson's correlation,

Supplementary Fig. 3b). This suggests that long-term protein oxidation negatively influences protein abundance, presumably due to proteasome-mediated protein degradation[37–39]. No correlation was observed between protein oxidation and phosphorylation with 2 h BCNU/AF treatment either globally ($r = 0.02$, $p = 0.45$ by Pearson's correlation, Supplementary Fig. 3a) or for annotated kinases and substrates ($r = 0.02$, $p = 0.76$ by Pearson's correlation, Supplementary Fig. 3b), suggesting changes in protein phosphorylation in response to BCNU/AF treatment were unlikely a direct result of oxidation of these proteins.

We hypothesised that global phosphorylation changes result from oxidative stress-induced changes in the activity of upstream kinases and/or phosphatases. To test this, we examined the crosstalk between oxidation and insulin signalling (Fig. 3a), specifically the kinases Akt, mTOR and AMPK. These kinases all had significantly altered levels of cysteine oxidation following BCNU/AF treatment (Fig. 3a, Supplementary Fig. 3c) and the phosphorylation of a large number of their known and predicted substrates[40] were altered upon BCNU/AF treatment (predominantly down-regulated) compared to controls (Fig. 3b, Supplementary Fig. 3d), consistent with the hypothesis that oxidation of these kinases regulates their activity.

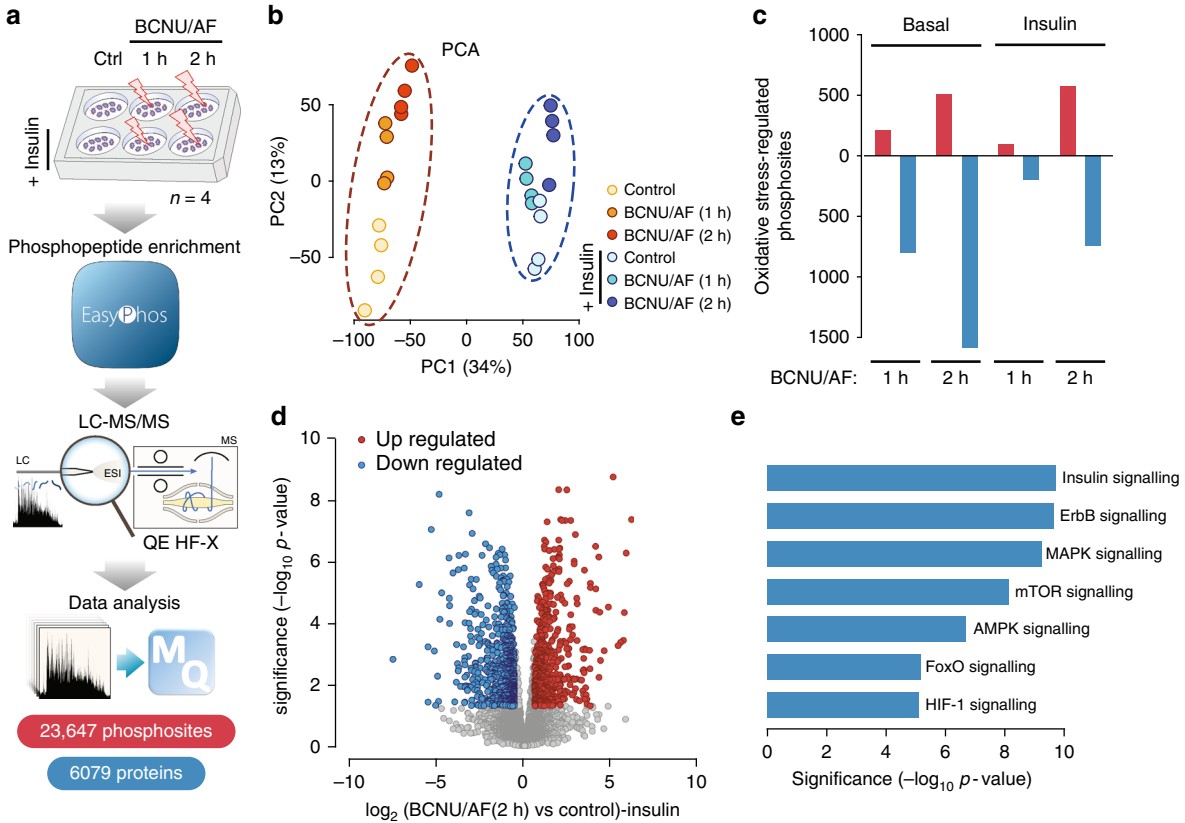

**Fig. 2** Oxidative stress-regulated phosphoproteome in adipocytes. **a** Experimental design of the phosphoproteomic analysis of insulin- and BCNU/AF-treated adipocytes. **b** Principal component analysis of the phosphoproteome. **c** Summary of oxidative stress-regulated phosphoproteome in adipocytes with or without insulin treatment. **d** Volcano plot showing oxidative stress-regulated phosphoproteome in adipocytes with 2 h BCNU/AF treatment in the presence of insulin. Significantly regulated phosphosites (by moderated $t$-test from limma R package) were indicated in red (up-regulated) or blue (down-regulated, adjusted $p < 0.05$ and absolute fold change > 1.5). **e** Gene ontology over-representation analysis (Fischer's exact test) of significantly oxidative stress-regulated phosphoproteins with 2 h BCNU/AF treatment in the absence of insulin.

Akt is activated by phosphorylation at two sites, T309 and S474, by the upstream kinases PDK1 and mTORC2, respectively. We observed oxidation of Akt at C77 (Fig. 3a and Supplementary Data 3) and hyper-phosphorylation of T309 under both basal and insulin-stimulated conditions (Fig. 3a, Supplementary Data 7) raising the possibility that oxidation may regulate Akt activation. BCNU/AF treatment had no effect on Akt S474 phosphorylation either with or without insulin stimulation (Supplementary Data 7). Akt was activated by insulin as indicated by increased phosphorylation of a range of its substrates (Fig. 3b, Supplementary Fig. 2g, Supplementary Data 7). However, insulin-stimulated phosphorylation of 11 Akt substrates was inhibited by oxidative stress (Fig. 3b, Supplementary Data 7) despite hyper-phosphorylation of Akt T309. While the reduction in phosphorylation of these Akt substrates was more pronounced after 2 h of BCNU/AF treatment (Fig. 3b, c), it was notable that not all Akt substrates were influenced by oxidative stress. This suggests that oxidative stress modifies Akt signalling in a substrate-specific manner rather than uniformly impairing the entire Akt signalling network.

Dysregulated substrate phosphorylation was even more pronounced for mTOR and AMPK (Fig. 3b, c), where we observed examples of substrate phosphorylation being inhibited or enhanced by oxidative stress. Since both kinases are regulated by Akt, these changes may in part reflect the reduced Akt activity under these conditions. However, both kinases were reversibly oxidized in the presence of BCNU/AF (mTOR on C713; AMPK on C205 on the gamma subunit), supporting the concept that

cysteine oxidation of these kinases may alter activity toward specific substrates.

Collectively, oxidative stress has a complex effect on kinase-substrate relationships and a loss of signal fidelity, as indicated by the increase in variability in substrate phosphorylation, is one of the most consistent outcomes of oxidative stress.

**Identification of oxidative modifications on Akt**. Our multi-omics analyses uncovered discordance between Akt phosphorylation at its activation site T309 and downstream substrate phosphorylation following acute oxidative stress. Given that we identified oxidation of C77 under the same conditions, we hypothesised that the direct oxidation of Akt itself could, at least in part, explain these observations. To explore this, we developed a non-reducing immunoprecipitation-mass spectrometry (IP-MS) assay to identify specific oxidative modifications of Akt (Supplementary Fig. 4a and b). Two disulfide bonds were identified in Akt2: C60-C77 within the PH domain (Supplementary Fig. 4c) and C297-C311 within the kinase domain (Supplementary Fig. 4d). We also observed two S-glutathionylation sites at C124 and C311 (Supplementary Fig. 4e and f). Importantly, these were exclusively identified in samples treated with BCNU/AF for 2 h, implying that the majority of Akt exists without oxidative modifications in the normal cellular redox environment. Both disulfides have been reported previously[41–43]. The disulfide between C297-C311 (the T-loop disulfide) has been implicated in kinase inhibition, possibly in combination with the oxidation of

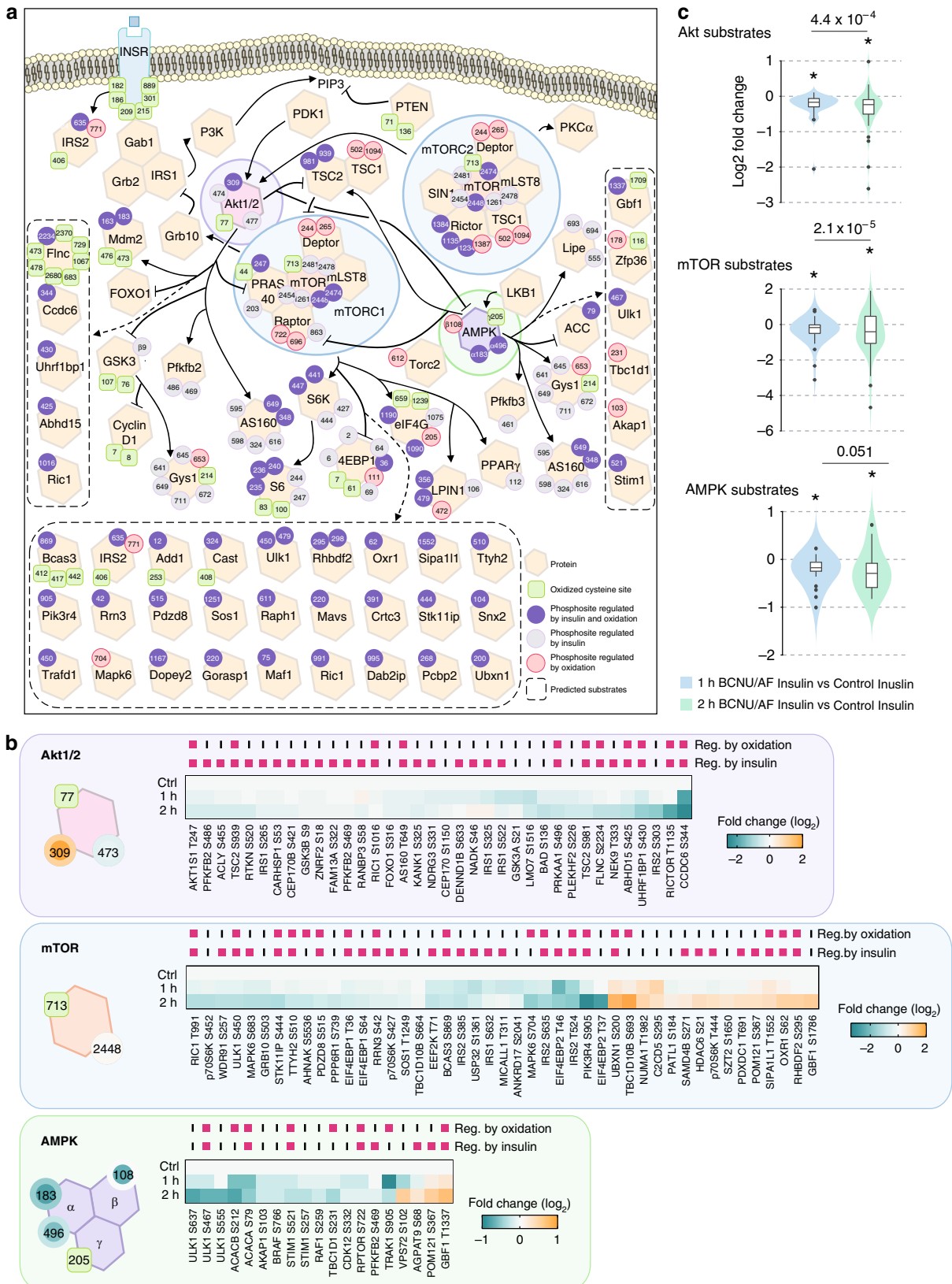

**Fig. 3** Integrative analysis of oxidative stress-regulated phospho-signalling networks. **a** A network diagram visualising the crosstalks of redox- and phospho-signalling. Oxidized cysteine sites (green squares) and phosphosites regulated by oxidation (red circles), insulin (grey circles) or both (purple circles) on key kinases and substrates of Akt, mTOR and AMPK signalling pathways were highlighted. **b** Heatmaps of standardised phosphorylation levels of Akt, mTOR, and AMPK substrates. Those that were significantly altered by oxidation and/or insulin were highlighted. **c** Boxplots for phosphorylation changes of Akt, mTOR and AMPK substrates with either 1 or 2 h BCNU/AF treatments vs controls in the presence of insulin. Boxes capture lower quartile and upper quartile with median displayed as a horizontal line in the middle; whiskers are min and max. A one-sided Bartlett Test was used to compare if the variance in (2 h BCNU/AF) was greater than (1 h BCNU/AF) treatment, following insulin stimulation. A one-sided $t$-test was used to compare if the mean was less than 0 ($p < 0.05$).

C124[41,42,44,45]. Whilst the functional consequence of the PH domain disulfide (C60-C77) has not been explored, it has been identified as a cross-strand disulfide, consistent with it acting as a disulfide switch that may regulate PIP3 binding[46].

**C60 and C77 are required for Akt activation**. The PH domain is essential for Akt recruitment to the PM and its phosphorylation at T309 and S474[47,48]. Since oxidative stress induced hyper-phosphorylation of Akt-T309, we hypothesized that the PH domain disulfide may promote Akt recruitment to the PM. To explore this and to avoid the confounding issue of endogenous Akt activity, we utilised the Akt-W80A MK2206-resistant system[49,50]. Treatment of cells with the Akt inhibitor MK2206 inhibited both endogenous Akt and overexpressed Akt2-WT activation as demonstrated by impaired phosphorylation of Akt at T309 and its substrates TSC2 and PRAS40 (Fig. 4a, b). Since the W80A mutation prevents binding of MK-2206 to Akt[49,50], insulin-stimulated activation of Akt2-W80A was not impaired by MK-2206 (Fig. 4a, b). Under these conditions, phosphorylation of Akt2-C60/77S-W80A at T309 and S474 and Akt substrates TSC2 and PRAS40 were impaired by approximately 70% (Fig. 4a, b).

Consistent with these observations, DNA content, cell growth and colony formation were lower in cells expressing Akt2-C60/77S-W80A compared to cells expressing Akt2-W80A (Fig. 4c–h) This implies that either C60, C77 or both Cys residues are necessary for insulin-stimulated Akt activation even in the absence of oxidative stress.

**C60 and C77 of Akt are essential for Akt activation in vivo**. To address whether C60 and C77 played a similar role in vivo we utilised the Gal4-UAS system[51] to overexpress human Akt2-WT (hAkt2-WT) or the Akt2-C60/77S mutant (hAkt2-C60/77S) in *Drosophila melanogaster*. We did not observe any gross abnormality between hAkt2-WT and hAkt2-C60/77S over-expressing flies, suggesting that endogenous dAkt1 function is sufficient to sustain normal growth and development. Overnight starvation followed by the administration of 10% glucose for 30 or 60 min increased phosphorylation of S505 on *Drosophila* Akt1 (dAkt1; Fig. 4i). Under the same conditions, hAkt2-WT, but not hAkt2-C60/77 S, displayed feeding-induced phosphorylation at T309 and S474, indicating that mutation of C60 and C77 inhibited Akt activation in vivo (Fig. 4i).

To determine if C60 and C77 of hAkt2 were functionally relevant in vivo we knocked down dAkt1, while overexpressing either hAkt2-WT or hAkt2-C60/77S. No live adults emerged with whole body knockdown of dAkt1, consistent with previous observations of early lethality[52]. In contrast, depletion of dAkt1 and expression of hAkt2-WT resulted in live adults (Fig. 4j). Flies expressing hAkt2-C60/77 S, were significantly smaller (Fig. 4j), and displayed decreased body weight, body length and wing size (Fig. 4k–m). hAkt2-WT, but not hAkt2-C60/77S, could be fully activated by refeeding, as determined by phosphorylation at T309 and S474 (Fig. 4n). Intriguingly, hAkt2-C60/77S was constitutively phosphorylated at T309, but S474 phosphorylation was impaired (Fig. 4n). Taken together, these data suggest C60 and C77 are required for Akt activation in vivo.

**MD analysis suggests C60 and C77 regulate PIP3 binding**. To gain mechanistic insight into the role of the PH domain disulfide, we performed molecular dynamics (MD) simulations using the PH domain of Akt with either the C60-C77 disulfide intact or where the two cysteine residues were substituted with serines. We focused this analysis on the 3 loops involved in phosphatidyli-nositol(3,4,5)-trisphosphate (PIP3) binding: loop 1 (residues 7–31), loop 2 (residues 52–66) and loop 3 (residues 72–90)

(Fig. 5a). Comparison of the snapshots at 100 ns with the reference crystal structure (PDB:1unp) revealed a conformational change at the tip of loop 1 (E17–Y18) in Akt-C60/77S but not in wild type (WT) Akt (Fig. 5b, c). Examination of the ionic bond between E17 and R86 revealed it was maintained in WT Akt but it was broken in Akt-C60/77S (Fig. 5d). The Cα–Cα distances between the same residues also increased in Akt-C60/77S (Fig. 5e), showing that this was not just due to the motion of the side chains but also involved loop 1 moving away from loop 3. Comparison of the RMSDs of loops 1, 2 and 3 between the WT and mutant Akt showed that breaking the disulfide bond had a large effect on the conformation of loop 1 but not on loops 2 and 3, which are connected by the disulfide bond (Fig. 5f–h). For more detailed information, we examined the residue-specific RMSDs in loop 1 (Fig. 5i), which revealed that the largest changes between WT Akt and Akt-C60/77S occurred around the E17 residue. For control purposes, we also performed MD simulations of Akt with protonated C60/77, which breaks the disulfide bond without affecting the side chains. Comparison of the residue-specific RMSDs in loop 1 with those of Akt-C60/77S showed similarly increased fluctuations in the vicinity of E17 (Fig. 5i, Supplementary Fig. 5. a–e), indicating that breaking of the disulfide bond is responsible for the changes in loop 1.

Since loop 1 residues contribute seven contacts to binding of PIP3 (loop 2 and 3 residues contribute only one contact each; Supplementary Table 1), the opening of the PIP3 binding pocket in the C60/77S mutant may impair its ability to bind PIP3. To test this, we docked PIP3 to WT Akt and Akt-C60/77S. The nine contacts obtained from docking of PIP3 to WT Akt were maintained during the MD simulations (Supplementary Table 1). In contrast, only three residues of Akt-C60/77S (K14, R25 and N53) were involved in docking of PIP3, which were rapidly broken in subsequent MD simulations (Fig. 5j–l) indicating that PIP3 binds less stably to Akt-C60/77S than WT Akt with the disulfide bond.

**C60 and C77 regulate PIP3 binding and Akt recruitment**. Consistent with our MD predictions, direct oxidation of the Akt PH domain with $H_2O_2$ increased its binding affinity for PIP3 (Fig. 6a, b). Total internal reflection fluorescence (TIRF) micro-scopy[53] of TagRFPt-Akt2-WT or TagRFPt-Akt2-C60/77S in live adipocytes revealed that insulin-stimulated translocation of the Akt2-C60/77S mutant to the cell surface was not detectable in the presence of either 1 or 100 nM insulin in contrast to Akt2-WT (Fig. 6c). This suggests that the impaired insulin-stimulated activation of Akt2-C60/77S (Fig. 4a) is due to impaired PM recruitment. Taken with the MD simulation, these data lead us to predict that: (1) the C60-C77 disulfide promotes Akt affinity for PIP3; (2) increased formation of the C60-C77 disulfide would promote Akt PM accumulation; and (3) that deletion of either cysteine residue would ablate these effects. If the disulfide bond in the PH domain of Akt is essential for recruitment, we would expect the single cysteine mutants to phenotypically mimic the response of the Akt2-C60/77S double mutant.

We measured insulin-stimulated PM translocation of Akt2-C60S and Akt2-C77S. Intriguingly, these mutants exhibited markedly different responses to insulin. Whilst the C77S mutant phenocopied the complete loss of recruitment observed for the C60/C77S mutant, the C60S mutant displayed an intermediate phenotype (Fig. 6c). Consistent with this, insulin stimulated phosphorylation of Akt2-C60S at T308 was five-fold lower than Akt WT and not detectable for either the C77S or C60/77S mutants (Fig. 6d, e).

These data suggest that both cysteines are necessary for normal insulin-stimulated Akt recruitment and that C77 plays a more significant role in insulin-stimulated Akt translocation than C60.

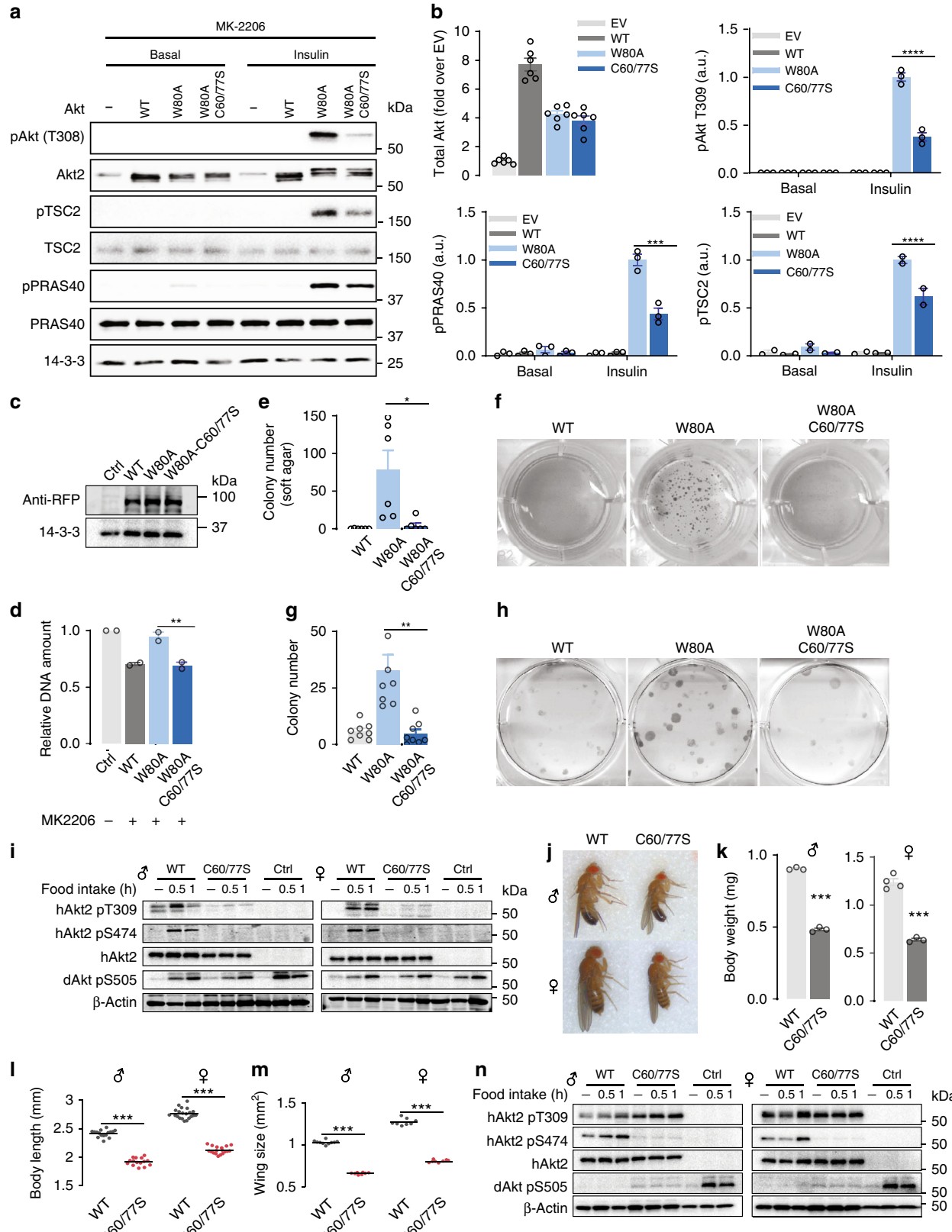

**Cancer mutation C77F functionally mimics oxidized PH domain**. The importance of C77 is reinforced by the activating breast cancer-associated Akt1-C77F mutation[54,55]. Indeed, both full length and PH domain Akt1-C77F mutants were hyper-responsive to insulin when compared to the WT controls

(Fig. 6f, g), suggesting that the Akt1-C77F mutation may functionally mimic oxidative modification and increase PIP3 binding.

MD simulations on the PH domain of Akt were performed to investigate how the C77F mutation could increase PIP3 affinity. Examination of the critical E17-R86 ionic bond revealed that

**Fig. 4** C60 and C77 of Akt are essential for Akt activation *in vitro and in vivo*. **a** Representative immunoblots of Akt activation and signalling. 3T3-L1 fibroblasts overexpressing Akt2 -WT, -W80A or -W80A-C60/77S were serum starved and then treated with MK-2206 (10 μM, 30 min) prior to stimulation with insulin. **b** Quantification of Akt activation and signalling blots. Data are mean ± SEM from $n = 3$, (***$p < 0.001$, ****$p < 0.0001$) by two-way ANOVA. **c** Expression level of Akt mutants in HeLa cells transfected with indicated vectors for 24 h. **d** Proliferation of HeLa cell overexpressing Akt2-WT or indicated mutants measured by Hoechst. Data are mean ± SEM from $n = 4$, **$p < 0.01$ by one-way ANOVA. **e** Colony formation assays were performed using HeLa cells overexpressing corresponding Akt2-WT or indicated mutants. Colony numbers were counted and shown as mean ± SEM from $n = 8$, *$p < 0.05$ by Brown-Forsythe and Welch ANOVA tests. **f** Representative images of soft HeLa colonies. **g** Anchorage-independent cell growth was assessed by soft agar assays using HEK cells overexpressing Akt2-WT or indicated mutants. Colony numbers were counted and shown as mean ± SEM from $n = 6$, *$p < 0.05$ by one-way ANOVA. **h** Representative images of HEK colonies. **i** Flies overexpressing hAkt2-WT or hAkt2-C60/77S were starved overnight and re-fed for 30 or 60 min. Flies were sacrificed and the samples were immunoblotted using indicated antibodies. **j** Representative images of flies with depletion of dAkt1 and simultaneous overexpression of human hAkt2-WT or hAkt2-C60/77S. **k-m** Fly growth were determined by the measurement of body weight (**k**), body length (**l**) and wing size (**m**); ***$p < 0.001$ by Welch's *t*-test. **n** Flies with depletion of dAkt1 and overexpression of hAkt2-WT or hAkt2-C60/77S were starved overnight and re-fed for 30 or 60 min. Flies were sacrificed and the samples were immunoblotted using indicated antibodies.

following a period of equilibration (~60 ns), the distance between the residues decreased and a highly stable state was achieved (Fig. 6h; Supplementary Fig. 5f–i). In WT Akt, E17 interacts with both K14 and R86, which is responsible for the fluctuations in the E17-R86 distance. The small conformational change caused by the C77F mutation brings loop 1 nearer to loop 3, enabling simultaneous contact of E17 with K14 and R86. This increased stability coincided with the formation of a hydrophobic pocket involving interactions between F77, F55, I75 and I84 (Fig. 6i, k). The residue-specific RMSDs in loop 1 were found to be similar to that of WT Akt (Fig. 5f). Thus, this hydrophobic interaction was as effective as the disulfide bond in stabilizing the PIP3 binding pocket. Indeed, when PIP3 was docked to Akt-C77F, a similar binding mode to that of WT Akt was obtained with nine contacts involving the same Akt residues (Supplementary Table 1). These contacts were maintained during the MD simulations as indicated by the average distances obtained from MD consistent with increased binding affinity when compared to C60/C77S and reduced WT Akt. This suggests that C77F functionally mimics oxidized Akt containing the C60-C77 disulfide, resulting in hyperactive Akt signalling.

**A role for the C60-C77 disulfide in growth factor signalling.** The above data are consistent with C60/C77 oxidation increasing PIP3 binding. Indeed, treatment of cells with BCNU/AF or $H_2O_2$ (through the addition of $H_2O_2$ directly, or produced by glucose oxidase in the medium) to oxidize Akt, increased insulin-stimulated PM recruitment of Akt (Fig. 7a, b).

Given that insulin has been reported to increase ROS[56–58], we propose a redox-dependent mechanism that promotes Akt translocation to the PM in response to growth factors, where growth factor-induced ROS, likely a product of NADPH oxidase 4 (NOX4) in adipocytes, oxidizes C60–C77 and enhances the Akt response to insulin-stimulated PIP3 production.

To explore this, we created a simplified mechanistic model of Akt signalling (Fig. 7c) and trained it with Akt PM recruitment (Fig. 7a) and Akt/Akt substrate phosphorylation data (Supplementary Fig. 6a, b) in the absence or presence of BCNU/AF (Detailed model description is given in the Supplementary Note 1 and Supplementary Tables 3, 4 and 5). The trained model quantitatively recapitulated the experimental data, lending support to the proposed mechanism (Fig. 7d). Next we examined how BCNU/AF would influence Akt binding to PIP3. Additionally, the model predicted enhanced PH domain PIP3 binding kinetics in the presence of BCNU/AF (Fig.7e), in close agreement with our observations of the purified PH domain binding to PIP3 (Fig. 6a, b). We then predicted the effect of NOX inhibition on insulin-stimulated Akt recruitment (Fig. 7f) and increasing concentrations of a NOX inhibitor (diphenylene iodinium, DPI) on the phosphorylation of Akt and its substrates (Fig. 7g). To test

the accuracy of the predicted response, we treated adipocytes with DPI prior to insulin stimulation. DPI impaired Akt PM recruitment at both 1 and 100 nM insulin (Fig. 7h). We also observed dose-dependent inhibition of Akt phosphorylation at T309, S473 and Akt substrate phosphorylation (PRAS40 T246 and Foxo1/3a T24/32) (Fig. 7i; Supplementary Fig. 6e–f). Despite the fact that DPI is known to bind to additional targets other than NOX4[59,60], these data are consistent with the model prediction (Fig. 7f).

This model assumes that insulin will increase formation of the PH domain disulfide, via the ROS burst. This was directly assessed in flag tagged Akt1 overexpressed in NIH/3T3 cells over a time course of insulin stimulation using differential cysteine labelling (Fig. 7j). Oxidation of C60 was observed in approximately 0.5% of the total Akt1 pool between 0 and 5 min of insulin stimulation (Fig. 7k), consistent with the highly reducing environment of the cytosol. The proportion of Akt that was oxidized at C60 increased 2.5-fold with longer term insulin stimulation (Fig. 7k; 10–60 min). The redox state of Cys297 of the kinase domain was unchanged with insulin stimulation (Fig. 7k). These data support the formation of the C60-C77 disulphide in Akt and that it is regulated by insulin under physiological conditions.

## Discussion

Protein oxidation is an efficient modulator of protein function under physiological and pathophysiological conditions. Here we combine a model of oxidative stress with a multi-omics approach to explore the interaction between ROS-dependent protein oxidation and phospho-signalling. ROS has a profound influence on the adipocyte phosphorylation signalling network principally by influencing the fidelity of signalling at key regulatory nodes including Akt, mTOR and AMPK. In particular, we identify a number of oxidative modifications including two disulfide bonds that form in Akt and provide evidence that oxidation of C60 and/ or C77, or the formation of a disulfide bond between these residues, stabilises the PIP3 binding pocket and promotes Akt activation through PM recruitment. Deletion of both C60 and C77 impairs Akt activity and results in cellular and whole-body phenotypes consistent with Akt hypoactivity.

Analysis of the redox proteome in combination with the phosphoproteome revealed a major effect of oxidative stress on signal transduction. To avoid the difficulties in interpreting unannotated and potentially promiscuous protein phosphorylation sites[61,62], we focused on three ROS-regulated kinases: Akt, mTOR and AMPK. Examination of kinase-substrate relationships in these subnetworks revealed varied effects of oxidation on different substrates (Fig. 3b, c). For example, only 11 out of 36 Akt substrates were modulated by treatment with BCNU/AF and the same was true for AMPK and mTOR, with these kinases

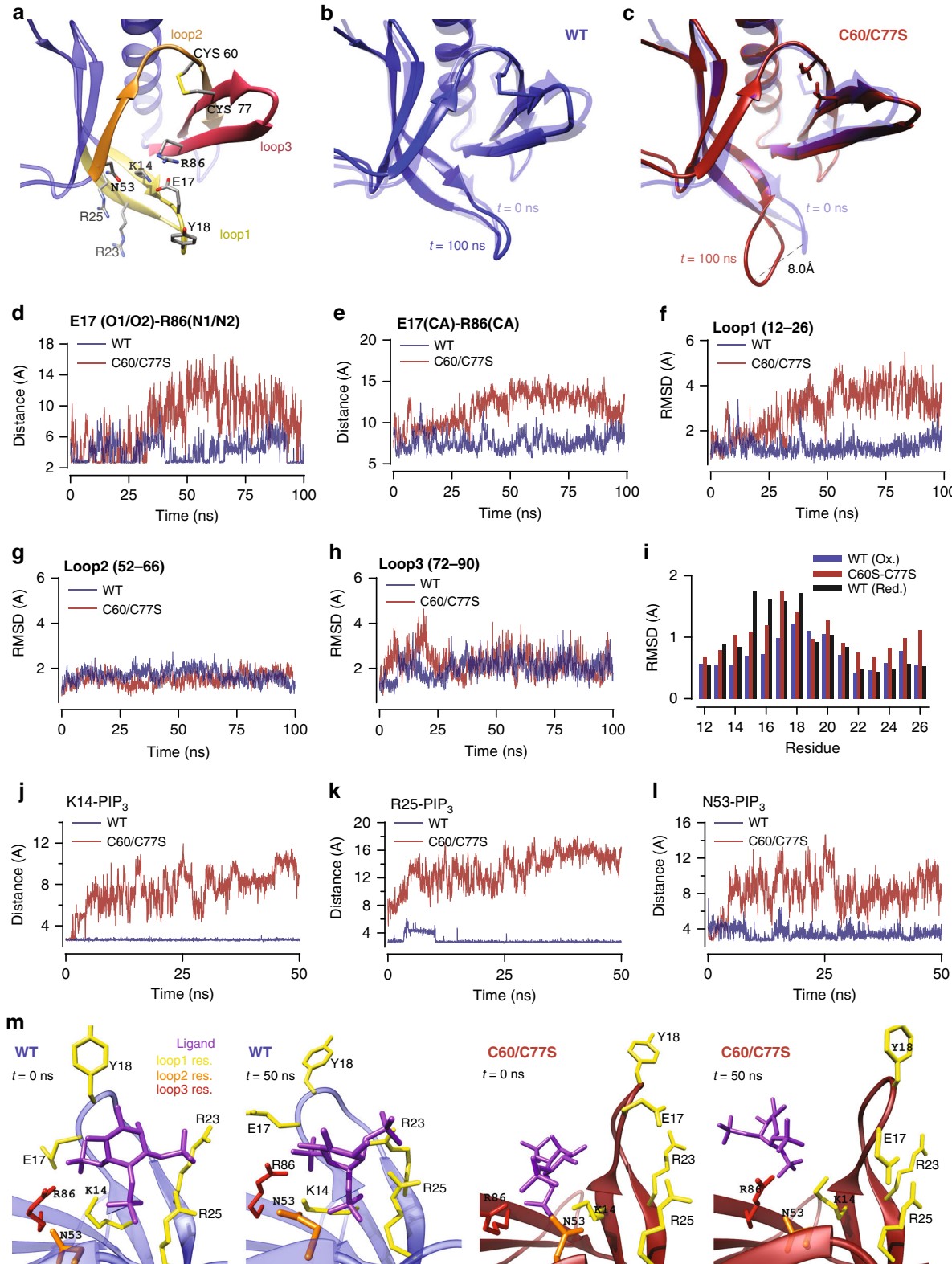

**Fig. 5** MD simulations predict the C60-C77 disulfide in Akt increases PIP3 affinity. **a** Akt1 PH domain (1unp) showing the disulfide bond between C60 and C77, the key residues required for PIP3 binding and loops 1 to 3. **b** Comparison of the Akt1 crystal structure 1unp (transparent) with the WT and **c** C60/C77S structures obtained after ~100 ns MD. **d** Time series of the E17(O)–R86(N) distance in WT and mutant Akt1. Only the distance between the closest O-N atoms are shown. **e** Time series of the Ca-Ca distances between the residues E17 in loop 1, and R86 in loop 3 in WT and mutant Akt1. **f–h** Comparison of the backbone RMSDs of the three loops in WT and mutant Akt1 calculated using the crystal structure as a reference. For loop 1, the RMSD is calculated for the residues 12–26, which are involved in binding of PIP3. **i** Residue specific RMSDs of the loop 1 residues 12–26 for the WT and C60/77S and protonated C60/77 mutant Akts. **j–l** Time series of the contact distances involved in PIP3 binding to WT Akt (blue) and Akt-C60/77/S (red). **j** WT Akt K14($N_Z$)–PIP3 (OPG/H) and Akt-C60/77S K14($N_Z$)–PIP3 (OPH). **k** WT Akt R25($N_2$)–PIP3 (O5P/O6P) and Akt-C60/77S R25($N_2$)–PIP3 (OPG). **l** WT Akt N53($N_{D2}$)–PIP3 (O6P/O8P) and Akt-C60/77S K14($N_{D2}$)–PIP3 (O8P). **m** Snapshots of WT (left-hand panels) and mutant (right-hand panels) Akt1-PIP3 complex at 0 and 50 ns.

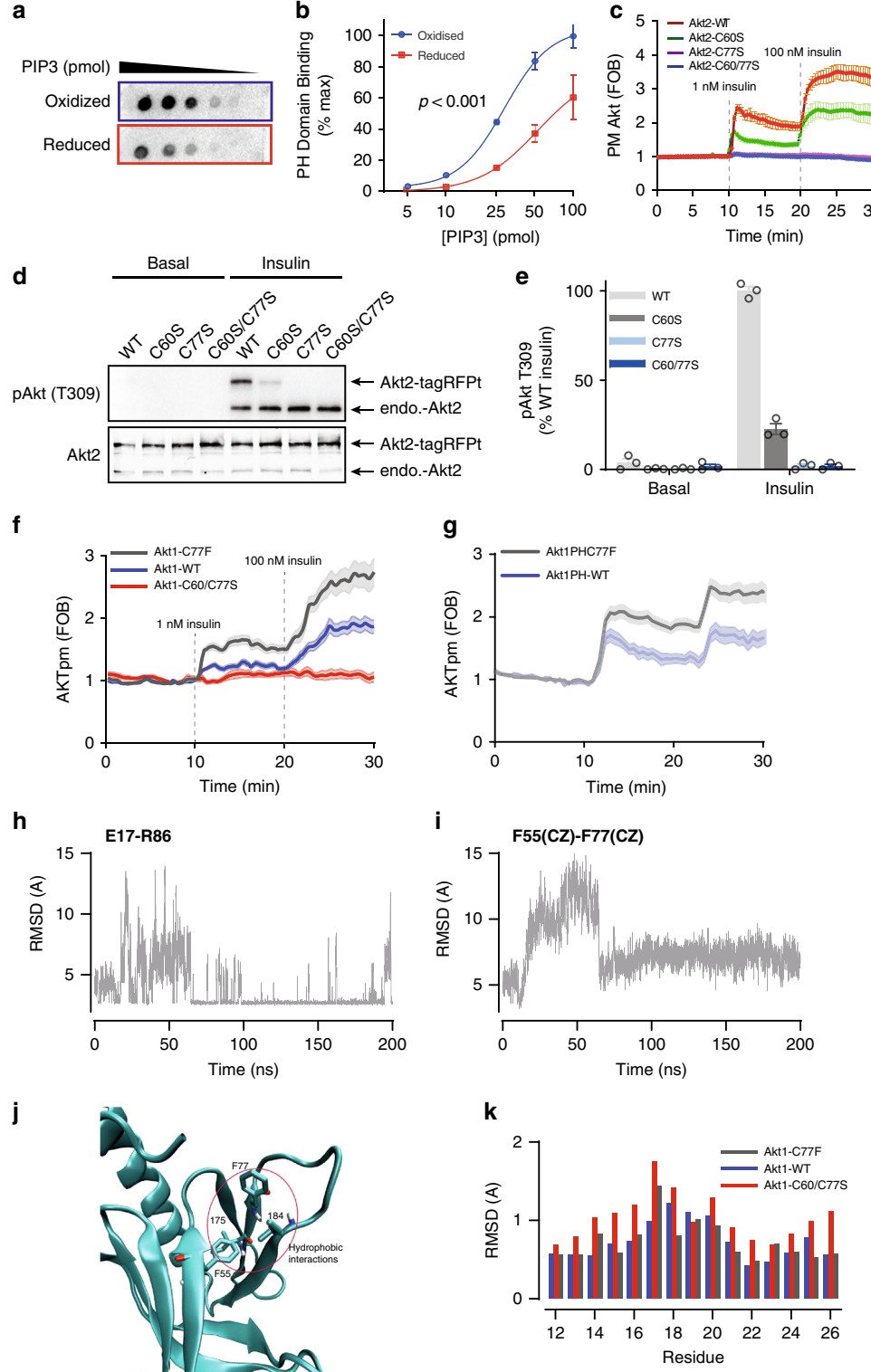

exhibiting both decreased and increased substrate phosphorylation. This suggests that oxidative stress impairs signal fidelity rather than kinase activity. This is perhaps not surprising because redox-dependent modifications of many intermediates in signalling pathways can have diverse functional outcomes such as alterations in activity, localisation or substrate specificity. In addition, oxidation of substrates may alter their localisation/binding partners and/or their cognate phosphatase activity, further exacerbating signal dysregulation. As such, the effect of

oxidative stress on the global phosphoproteome will represent the integrated output of all of these influences. Therefore, we propose that oxidative stress will induce greater variability in the phosphorylation of substrates for kinases with a greater number of regulatory inputs. Consistent with this idea, substrate dysregulation was most marked for mTOR, which is downstream of multiple regulatory inputs including Akt.

Detailed analysis of Akt revealed how the integrated effect of different oxidative modifications within the one kinase can define

**Fig. 6** C60 and C77 are required for Akt activation. **a** In vitro protein-lipid overlay assay. Aliquots of purified Akt2 PH domain (N-terminal 111 residues) was reduced or oxidized by DTT or $H_2O_2$, respectively, and then incubated with membranes spotted with serial dilutions of PIP3. Binding affinity was assessed by immunoblotting. **b** Quantification of in vitro protein-lipid overlay assay. Data are shown as mean ± SEM from $n = 3$. For oxidized Akt2 PH domain, Hillslope = 2.407, EC50 = 28.84; for reduced Akt2 PH domain, Hillslope = 1.908, EC50 = 52.30. $p$-value from two-way ANOVA. **c** Assessment of Akt translocation to the PM by TIRF. Adipocytes overexpressing TagRFPt-Akt2-WT or TagRFPt-Akt2-C60/77S were serum starved, followed by insulin treatments (1 nM and 100 nM). Data are shown as mean ± SEM. **d** Representative immunoblots of Akt2 phosphorylation at T309. 3T3-L1 fibroblasts overexpressing TagRFPt tagged Akt2 -WT, C60S, C77S or C60/77S were serum starved and then treated with MK-2206 (10 μM, 30 min) prior to stimulation with insulin. **e** Quantification of T309 phosphorylation. Data is mean ± SEM from $n = 3$. **f** Translocation of tagRFPt tagged Akt1 WT, C77F or C60/C77S to the plasma membrane in response to 1 nM or 100 nM insulin, measured by TIRFm. **g, f** Translocation of tagRFPt tagged Akt1 WT, C77F or C60/C77S PH domains to the plasma membrane in response to 1 nM or 100 nM insulin, measured by TIRFm. **h** Time series of the E17(O)-R86(N) distance in Akt1-C77F. Only the distance between the closest O–N atoms are shown. **i** Time series of the F55(CZ)-F77(CD1) distance in the C77F mutant Akt1. Equilibrium configuration is reached after 60 ns. **j** C77F hydrophobic interactions. The F77 side chain strongly interacts with the F55, I75 and I84 residues which form a hydrophobic pocket. **k** Residue specific RMSDs of the loop 1 residues 12-26 for the WT and C60/77S and protonated C60/77 mutant Akts.

signalling outcomes. Akt is hyper-phosphorylated at its activating site T309 under oxidative stress and yet its activity appears impaired (Fig. 3b, Supplementary Fig. 3c and Supplementary Fig. 6a). We propose that Akt hyper-phosphorylation is driven (at least in part) by increased Akt PM recruitment that is supported by oxidation of C60–C77, while Akt kinase activity is attenuated by oxidation of C297–C311 perhaps in combination with glutathionylation of C124, as has been described previously[41,42,44,45]. We note that BCNU/AF also increased both PM localization and Akt T308 phosphorylation in the absence of insulin. The magnitude of the basal T308 increase combined with the modest increase in PM localised Akt, leads us to speculate that additional mechanisms may be involved in the hyperphosphorylation. For example the C297–C311 disulfide may protect T308 from phosphatases or the T308 phosphatase itself (PP2B) may be impaired by ROS as ROS has been shown to impair Ser/Thr phosphatase activity[63].

The PH domain disulfide (C60–C77), first identified in crystal structures[43], is a cross strand disulfide (CSD) that has been conserved since its appearance prior to the triplication of Akt[46]. It has the archetypal –RH staple allosteric configuration, suggesting that it is redox active[46,64,65]. Mutational analysis of Akt C60 and C77 implies that these residues are required for the PIP3-dependent recruitment of Akt to the PM and this is supported by MD simulations that suggest that the presence of the disulfide bond between these residues stabilises the PIP3 binding pocket.

Our data suggest that C77 may play a more central role in Akt activation than C60. One possibility is that C77 oxidation alone, without formation of a disulfide bond with C60, can regulate Akt activation to some extent. The critical role of this residue in Akt activation is exemplified by the oncogenic mutation C77F, which increases Akt activity due to increased affinity for PIP3. Mechanistically this occurs via the formation of strong hydrophobic interactions between F77 and the hydrophobic pocket (F55, I75 and I84), that stabilises PIP3 binding. Unlike the disulfide bond, this interaction is not reversible and results in hyperactive Akt. Although we utilised a model of oxidative stress to initially identify the C60–C77 disulfide bond in Akt, we also demonstrate increased C60–C77 bond abundance following insulin-stimulation (Fig. 7k), suggesting that this site is sensitive physiological ROS. Therefore, we propose a redox-dependent mechanism for promoting Akt translocation to the PM in response to growth factors (Fig. 8). Previous studies have demonstrated that growth factors, such as insulin, stimulate a localised burst of ROS (likely at the PM) that is essential for signal transduction[56–58]. Our data suggest a model where insulin increases the disulfide between C60–C77, increasing the affinity of Akt for PIP3 and promoting Akt activation. We note that under severe oxidative stress, accumulation of Akt at the PM may be exacerbated by the suppression of PTEN activity, resulting in increased PIP3[6]. Whilst our analysis has focused on the PH domain, it has been proposed that the PH domain auto-inhibits the kinase domain, which is relieved when the PH domain binds membrane lipid[66]. Thus it is possible that the redox state of the PH domain cysteines may also regulate the PH domain:kinase domain interaction.

We have recently shown that Akt is first phosphorylated at T309 leading to partial activation of the kinase[67]. Akt then phosphorylates and activates the mTORC2 complex in a positive feedback loop that facilitates phosphorylation of Akt at S474 to fully activate the kinase[50,68]. We observed uncoupling of Akt T309 and S474 phosphorylation in BCNU/AF-treated cells (Supplementary Data 7), likely due to the formation of an inhibitory disulfide bond (C297–C311) in the kinase domain. This would impair mTORC2 activation and decrease S474 phosphorylation. Indeed it has been demonstrated that loss of the hydrophobic motif phosphorylation site in *Drosophila*, by either mutation or truncation, has no effect on fly growth, whilst loss of TORC2 activity results in impaired PIP3/Akt-dependent hyperplasia[69]. Thus, the observed growth phenotype in our *Drosophila* experiments is likely a result of lower TORC2 activity, downstream of impaired Akt activation.

Our multi-omics approach has produced a comprehensive atlas integrating global quantitative changes to the redox proteome, phosphoproteome and proteome of adipocytes that serves as a resource for research into the interaction between redox and phospho-signalling. Our analysis reveals widespread and complex crosstalk between these pathways that culminates in a loss of signal fidelity, and highlights a regulatory mechanism on Akt.

## Methods

**Antibodies used in this study**. anti-PRDX2 (Abcam, ab71533); anti-PRDX3 (Abfrontier, BS-1874R); anti-AKT-pT309 (CST, 9275); anti-AKT-pS474 (CST, 4051); anti-AKT2 (CST, 3063); anti-AS160-pT642 (CST, 4288); anti-TSC2-pT1462 (CST, 3611); anti-PRAS40-pT247 (CST, 2997); anti-GSK3a-pS21 (CST, 9327); anti-IRS1-pS307 (CST, 2381); anti-FOXO1-pT256 (CST, 9461); anti-mTOR-pS2448 (CST, 5336); anti-p70S6K-pT390 (CST, 9234); anti-p70S6K-pT421/S424 (CST, 9204); anti-RICTOR-pT1135 (CST, 3806); anti-S6-pS240/S244 (CST, 2215); anti-AMPKa-pT183 (CST, 2535); anti-ACC-pS79 (CST, 3661); anti-dAKT-pS505 (CST, 4054); anti-Tubulin (CST, 5335); anti-Tubulin (CST, 5335); anti-Actin (CST, 4967); anti-RFP (Life technologies, R10367); anti-FLAG (Sigma, F3165); anti-14-3-3 (Santa Cruz, sc-1657); all the other antibodies are home made.

**Cell culture and treatments**. 3T3-L1 fibroblasts (a gift from Howard Green, Harvard Medical School) were grown in Dulbecco's modified Eagle's medium (DMEM) containing 10% fetal bovine serum (FBS, Hyclone Laboratories) and 2 mM GlutaMAX (Gibco) in 10% CO2 at 37 °C. For differentiation into adipocytes, cells were grown to confluence, then treated with DMEM/FBS containing 4 μg/ml insulin, 0.25 mM dexamethasone, 0.5 mM 3-isobutyl-1-methylxanthine and 100 ng/ml d-biotin. After 72 h, the differentiation medium was replaced with fresh FCS/DMEM containing 4 μg/ml insulin and 100 ng/ml d-biotin for a further 3 days, then replaced with fresh FCS/DMEM. Adipocytes were re-fed with FCS/DMEM every 48 h and utilized for experiments between 9 and 12 days after the initiation of differentiation. These cells were routinely tested for mycoplasma.

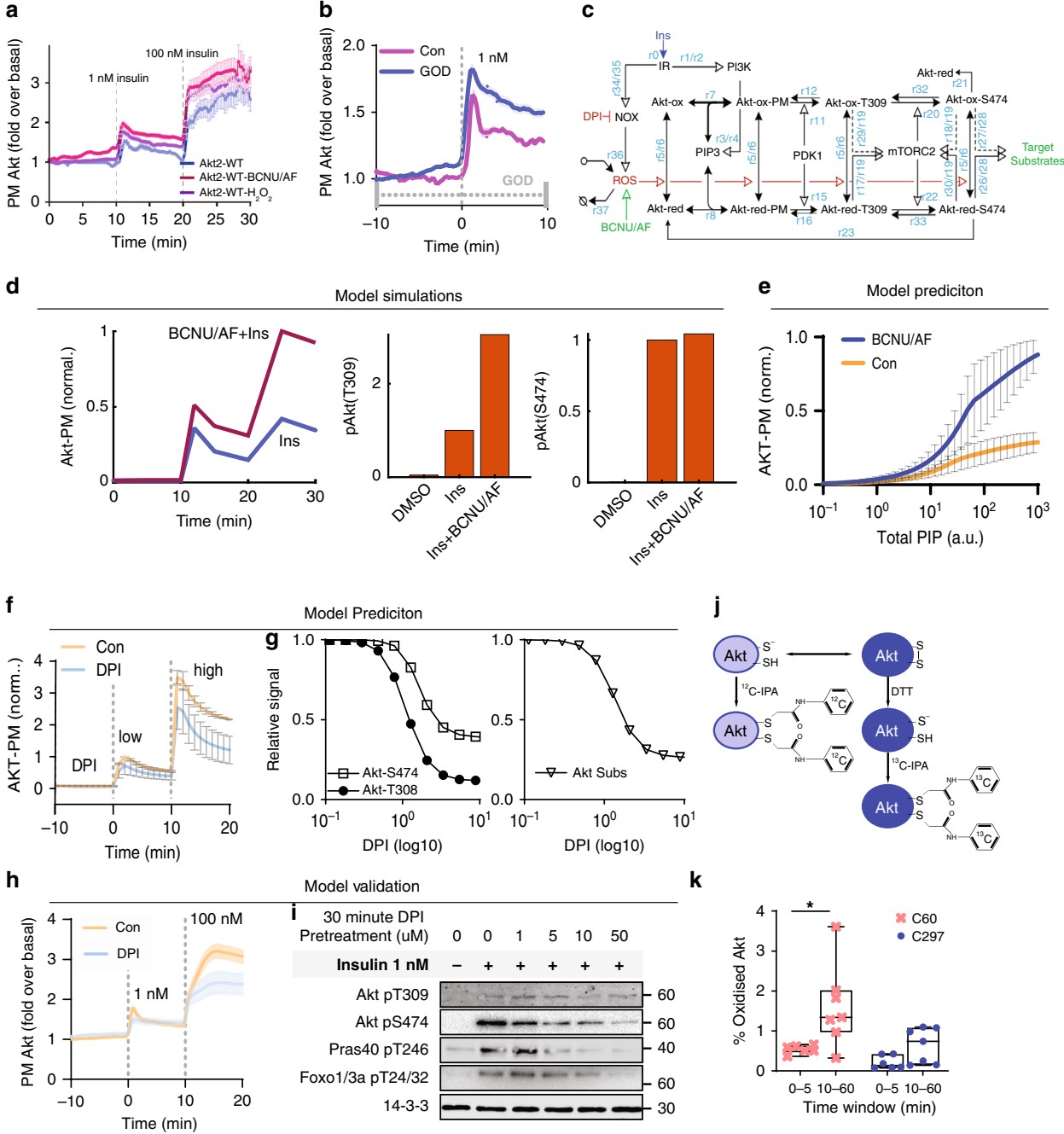

**Fig. 7** A physiological role for C60 and C77. **a**, **b** TIRF imaging of adipocytes overexpressing TagRFPt-Akt2-WT treated with **a** BCNU/AF or H₂O₂ (100 μM) or **b** glucose oxidase (GOD) as indicated prior to stimulation with 1 nM insulin. Data are mean ± SEM. **c** The simplified mechanistic AKT oxidation-phosphorylation network model. Ins Insulin, IR Insulin receptor, NOX NADPH oxidase 4, DPI diphenylene iodinium, ROS reactive oxygen species, Akt-ox oxidized Akt, Akt-red reduced Akt, Akt-ox-PM oxidized PM Akt, Akt-red-PM reduced PM Akt. Reaction numbers rx ($x = 0$–37) denote the model reactions (Supplementary Tables 1 and 2). **d** Simulated response of Akt-PM and phospho-Akt levels with either insulin stimulation alone, or in combination with BCNU/AF, using a representative best-fitted model parameter set. **e** Model prediction of the influence of BCNU/AF on Akt recruitment in response to increasing amounts of PIP3. Predictions are averaged using three independent best-fitted parameter sets (Supplementary Table 2); Data are mean ± SEM. **f** Model prediction of the effect of DPI on Akt recruitment under a low (at 0 min time-point) or high level (at 10 min time-point) of insulin stimulation. Data are mean ± SEM. **g** Model prediction of the effect of increasing DPI concentration on Akt (left panel) and Akt substrate phosphorylation (right panel). DPI was added for 30 min prior to a 10 min insulin stimulus. **h**, **i** Experimental validation of model predictions. **h** Adipocytes overexpressing TagRFP-T-Akt2-WT were imaged by TIRFm and treated with either DMSO or 10 μM DPI 10 min prior to insulin stimulation. **i** DPI dose response. Adipocytes were pretreated with the indicated dose of DPI for 30 min prior to stimulation with 1 nM insulin. **j** Differential cysteine alkylation and mass spectrometry method of measuring the redox state of Akt1 cysteines. Unpaired cysteine thiols were alkylated with 12C-IPA and the oxidized thiols with 13C-IPA following reduction with DTT. **k** Box (inter-quartile range; median) and whiskers (min:max) plots showing the oxidation of Cys60 and Cys297 in NIH/3T3 fibroblasts in the presence or absence of 10 nM insulin. Peptide were binned into 0–5 min and 10–60 min insulin periods. Two biological replicates were performed. *$p < 0.05$ by two-sided Wilcoxon rank sum test.

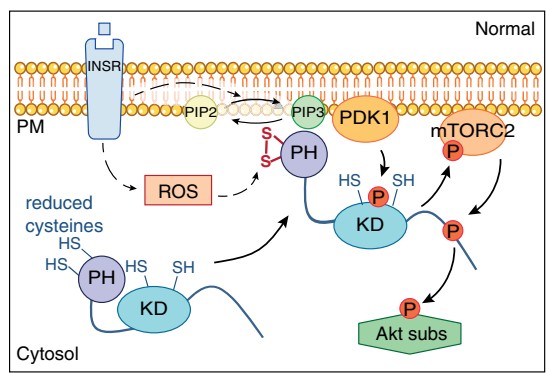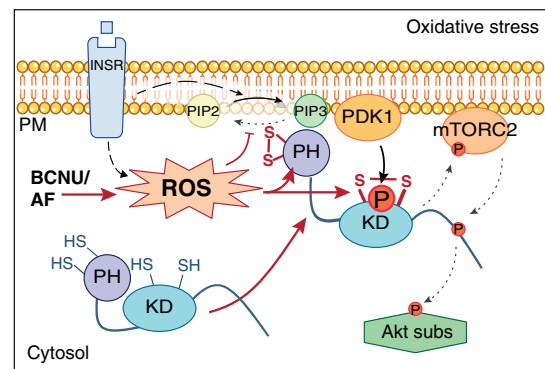

**Fig. 8** Model depicting redox-dependent mechanism for Akt activation. Insulin stimulation promotes an increase in cytosolic ROS (possibly via NOX4). This oxidizes the PH domain cysteines and augments the recruitment of Akt to the PM. Under condition of oxidative stress, the PH domain gets hyper-oxidized, which in concert with other ROS effects, such as the inhibition of PTEN, enhances Akt recruitment and leads to hyperphosphorylation of T308 in Akt. Under these conditions, an inhibitory disulphide (Cys297–Cys311) also forms in the kinase domain. This reduces Akt activity resulting in impaired substrate (Akt subs) phosphorylation. Dotted lines denote multiple steps between nodes. Solid lines denote a direct connection.

For total proteomic analysis, 3T3-L1 fibroblasts were triple SILAC labelled[70]. All studies used adipocytes between 8 and 12 days after differentiation. Cells were treated with 100 µM BCNU or 1 µM AF for the indicated times.

L6 rat myoblasts (ATCC® CRL-1458™) were grown in α-Minimum essential medium (α-MEM) containing 10% FBS (Hyclone Laboratories) in 10% CO2 at 37 °C. L6 myoblasts were differentiated into myotubes using α-MEM with 2% FBS when myoblasts reached 90–95% confluence. All L6 studies used myotubes between 6 and 8 days after differentiation.

**Plasmid transfection**. HEK293 (ATCC® CRL-1573™) or HeLa (ATCC® CCL-2™) cells were cultured in DMEM containing 10% FBS and 2 mM GlutaMAX in 10% CO2 at 37 °C, which were transfected at approximately 80% confluency using Lipofectamine 2000 (Life Technologies) according to the manufacturer's instructions.

**Sample preparation and mass spectrometry of redox proteome**. Adipocytes were incubated with ice-cold PBS containing 10% TCA for 1 min on ice, and washed once. Cells were then harvested in buffer containing 50 mM HEPES, 1 mM EDTA, 0.1% SDS and 10% TCA. Proteins were precipitated at 4 °C for 2 h. Samples were then centrifuged at $16,000 \times g$ for 20 min at 4 °C, and the pellets were washed once using pre-chilled acetone. Protein pellets were resuspended in buffer containing 6 M Urea, 50 mM HEPES (pH 8.5) and alkylated with 100 mM IAA at room temperature for 40 min in the dark. Excess IAA was removed by using 30 k filters and proteins were re-suspend in HES buffer (50 mM HEPES, 1 mM EDTA, 0.1% SDS, pH 8.0). Protein concentrations were determined by BCA protein assay and adjusted to 1 µg/µL. To each 100 µg sample, 5 mM TCEP was used for reducing at 50 °C for 1 h, and then iodoTMT quantitative tags were added and incubated at 37 °C for another 1 h in darkness. After adding 20 mM DTT to quench the reaction, equal amounts of each labelled sample were combined and precipitated using pre-chilled acetone overnight. Protein pellets were resuspended in buffer containing 6 M urea, 2 M thiourea, 25 mM TEAB, and protein mixture was digested using LysC (enzyme:substrate 1:100) at 30 °C for 2 h. The protein solution was then diluted 1:5 using 25 mM TEAB and further digested with trypsin (enzyme:substrate 1:50) at 37 °C overnight. The peptide mixture was desalted using stage-tips and separated by HILIC system for mass spectrometry analysis. Four replicates were performed including two forward labelling and two reverse labelling strategies. For the forward labelling, tag-126, -127, -128, -129, -130 and -131 were used to label the samples Ctrl1, BCNU 2 h, BCNU 24 h, Ctrl2, BCNU/AF 2 h and BCNU/AF 24 h, respectively. For the reverse labelling, tag-126, -127, -128, -129, -130 and -131 were used to label BCNU/AF 24 h, BCNU/AF 2 h, Ctrl2, BCNU 24 h, BCNU 2 h and Ctrl1, respectively.

MS-based redox proteomic analysis was performed on an Easy nLC-1000 UHPLC coupled to a Q Exactive mass spectrometer in positive polarity mode. Peptides were separated using an in-house packed 75 µm × 40 cm column (1.9 µm particle size, ReproSil Pur C18-AQ) with a gradient of 5–30% ACN containing 0.1% FA over 90 min at 200 nL/min at 55 °C. The MS1 scan was acquired from 300 to 1750 $m/z$ (70,000 resolution, 3e6 AGC, 100 ms injection time) followed by MS/MS data-dependent acquisition of the top 20 ions with HCD (17,500 resolution, 5e5 AGC, 60 ms injection time, 30 NCE, 2.0 $m/z$ isolation width).

Raw data was processed using MaxQuant (v1.6.1.2)[71] against a UniProt mouse database (06/2017, 59,581 entries) with default settings and minor changes: Oxidation of methionine and acetylation of protein N-terminus were set as variable modifications. IodoTMT6-plex of isobaric labels was enabled and reporter mass tolerance was set to 0.01 Da. Second peptides search was enabled. Peptide match tolerance was set to 20 ppm and 4.5 pm for first and main searches respectively and MS/MS match tolerance set to 20 ppm. Both peptide spectral match (PSM) and

protein FDR were set to 1%. This led to the identification of 281,345 unmodified and 66,002 iodoTMT-modified PSMs.

**Sample preparation and mass spectrometry of phosphoproteome**. Samples selected for MS analysis were processed using the recently described EasyPhos workflow[35]. Briefly, cells were washed three times with TBS, scraped in SDC lysis buffer (4% (w/v) SDC, 100 mM Tris HCl, pH 8.5) and boiled immediately (95 °C, 5 min). Lysates were cooled on ice, sonicated ($2 \times 30$ s), and centrifuged at $20,000 \times g$ for 15 min at 4 °C until a layer of fat formed. Clarified protein was carefully collected without disturbing the upper fat layer, and protein concentration was determined by BCA assay. Four hundred micrograms of protein was reduced and alkylated in a single step using TCEP and 2-Chloroacetamide respectively, and protein was digested in a 96-well plate by the addition of Trypsin and LysC (1:100 enzyme-protein ratio) overnight at 37 °C. Phosphopeptides were enriched in parallel format using the EasyPhos workflow as described[36]. To the digested peptides in 300 µl of SDC buffer (4% sodium deoxycholate, 100 mM Tris pH 8.5) in a 2 mL Deep Well Plate, 400 µl isopropanol was added and samples mixed (30 sec, 1,500 rpm). 100 µl enrichment buffer (48% TFA, 8 mM KH$_2$PO$_4$) was added and samples mixed (30 sec, 1,500 rpm). TiO$_2$ beads were subsequently added to the peptides at a ratio of 12:1 beads/protein (suspended in 80% acetonitrile, 6% TFA) and incubated at 40 °C for 5 min at 2000 r.p.m. Beads were subsequently pelleted by centrifugation ($2000 \times g$, 1 min) and the supernatant (containing non-phosphopeptides) was aspirated and discarded. Beads were re-suspended in Wash buffer (60% isopropanol, 5% TFA) with mixing (2000 rpm, 30 s), then pelleted ($2000 \times g$, 1 min) and the supernatant discarded. Beads were washed a further four additional times with 1 ml wash buffer. After the final wash, beads were re-suspended in 100 µl transfer buffer (60% ACN, 0.1% TFA) and transferred onto the top of a C8 StageTip, and centrifuged for 3–5 min at $500 \times g$ until no liquid remained on StageTip. Bound phosphopeptides were eluted 2× with 30 µl Elution buffer (40% ACN, 20% NH4OH (25%, HPLC grade)), and collected by centrifugation into clean PCR tubes. Samples were concentrated in a SpeedVac for 30 min at 45 °C and then immediately acidified by the addition of 99% isopropanol/1% TFA, and loaded directly onto the top of an SDB-RPS StageTip. After loading StageTips were washed once with SDB-RPS Wash buffer 1 (99% isopropanol/1% TFA) and once with SDB-RPS Wash buffer 2 (0.2% TFA, 5% acetonitrile), then phosphopeptides were eluted from StageTips with 60 µl of SDB-RPS elution buffer (60% acetonitrile, 0.5% NH$_4$OH (25%, HPLC grade)). Samples were concentrated in a SpeedVac for ~45 min, at 45 °C, then resuspended in 7 µl MS loading buffer (0.3% TFA, 2% acetonitrile) for mass spectrometry analysis.

We applied a label-free quantification approach and measured samples in a single-run format on a benchtop Orbitrap (Q Exactive HF-X) mass spectrometer[72], and all phosphoproteomics experiments were performed with four biological replicates. Peptides were separated using an in-house packed 75 µm × 40 cm column (1.9 µm particle size, ReproSil Pur C18-AQ) with a gradient of 3-41% ACN containing 0.1% FA over 90 min at 350 nL/min at 55 °C. The MS1 scan was acquired from 300–1600 $m/z$ (60,000 resolution, 3e6 AGC, 120 ms injection time) followed by MS/MS data-dependent acquisition of the top 10 ions with HCD (15,000 resolution, 1e5 AGC, 50 ms injection time, 27 NCE, 1.6 $m/z$ isolation width).

Raw data was processed using MaxQuant (v1.6.1.0) against the UniProt mouse database (07/2017, 59,609 entries). Oxidation of methionine, acetylation of protein N-terminus and phosphorylation of serine, threonine and tyrosine were set as variable modifications. Carbamidomethylation of cysteine was set as a fixed modification. Second peptides and match between runs options were enabled, with the matching time window set to 1 min. FDR at the PSM, protein, and site levels were each 1%.

**Sample preparation and mass spectrometry of total proteome.** For the total proteome study, adipocytes were triple SILAC labelled. Briefly, adipocytes were washed with ice-cold PBS and harvested in lysis buffer containing 6 M urea, 2 M thiourea, 25 mM TEAB, 0.1% SDS. After sonication, cell lysates were centrifuged at 15,000 g for 30 min at room temperature. The supernatant was precipitated using pre-chilled acetone overnight. Protein pellets were resuspended in buffer containing 6 M urea, 2 M thiourea, 25 mM TEAB. After reduction with 10 mM DTT at room temperature for 60 min and alkylation with 25 mM IAA at room temperature for 30 min in the dark, the protein mixture was digested using LysC (enzyme: substrate 1:100) at 30 °C for 2 h. The protein solution was then diluted 1:5 using 25 mM TEAB and further digested with trypsin (enzyme:substrate 1:50) at 37 °C overnight. The peptide mixture was desalted using stage-tips and separated by HILIC system for mass spectrometry analysis. In this study, three biological replicates were performed.

Mass spectrometry analysis was performed on the same LC/MS system including an Easy nLC-1000 UHPLC coupled to a Q Exactive mass spectrometer with the same method as redox proteomic analysis except that NCE was set to 27.

Raw data was processed using MaxQuant (v1.6.1.2) against the same UniProt mouse database (06/2017, 59,581 entries). Oxidation of methionine and acetylation of protein N-terminus were set as variable modifications. Carbamidomethylation of cysteine was set as a fixed modification, as well as triple SILAC labels (Lys0/Arg0, Lys4/Arg6, and Lys8/Arg10). The re-quantify, second peptides search and match between runs options were enabled. Unique and razor peptides were used for protein quantification and the minimum ratio count was set to 2. Peptide match tolerance was set to 20 ppm and 4.5 pm for first and main searches respectively and MS/MS match tolerance set to 20 ppm. Both PSM and protein FDR were set to 1%.

**Identification of redox PTMs on Akt by IP-MS.** HEK293 cells were transfected with FLAG-Akt2 using Lipofectamine 2000 and all studies were performed 48 h post-transfection. Prior to IP, transfected HEK293 cells were incubated with or without 100 µM BCNU and 1 µM auranofin for 2 h. Overexpressed FLAG-Akt2 protein was purified as described previously[73]. To retain endogenous oxidation, no reducing reagent was used during sample processing. In addition, the pH was adjusted to 6.5 to minimise artificial oxidation[74]. Briefly, cells were washed with ice-cold PBS once and lysed in 1 mL of NP-40 buffer (1% NP-40, 10% glycerol, 150 mM NaCl, 50 mM Tris-HCl, pH 6.5) using a 22-gauge needle six times and then a 27-gauge needle three times followed by centrifugation at 16,000 × g for 10 min at 4 °C to remove cellular debris. Each cell lysate was incubated with 40 µL slurry of protein G beads (GE Life Sciences) and 1 µL of anti-FLAG antibody (Sigma-Aldrich; F3165) for 2 h at 4 °C with rotation. The beads were washed three times with NP-40 buffer and twice with PBS (pH 6.5), and then were resuspended in 40 µL of PBS (pH 6.5) containing 3xFLAG peptide (0.2 µg/µL, Sigma-Aldrich) to elute FLAG-Akt2.

The purified FLAG-Akt2 solution was diluted using 25 mM TEAB and digested with 1 µg trypsin at 37 °C overnight. The peptide mixture was desalted using stage-tips. In this study, three replicates were performed.

Mass spectrometry analysis was performed on the same LC/MS system including an Easy nLC-1000 UHPLC coupled to a Q Exactive mass spectrometer with the same method as redox proteomic analysis except that NCE was set to 25.

Disulfide bonds were identified using pLink-SS as described previously[74]. Five missed cleavages for trypsin digestion were allowed. Oxidation of methionine, acetylation of protein N-terminus and Gln to pyro-Glu were set as variable modifications. Search results were filtered by requiring precursor tolerance < 10 ppm. Only human Akt2 protein sequence was used for searching, and all identified MS2 spectra were manually checked and labelled.

To identify the other reversible oxidative modifications on cysteines, raw data was processed using MaxQuant (v1.5.2.10) against a UniProt human database (09/2014) with default setting of 4.5 ppm and 20 ppm for main search precursor and fragment ions tolerance, respectively. Oxidation of methionine and acetylation of protein N-terminus were set as variable modifications, as well as S-Acylation, S-Glutathionylation, S-Nitrosylation, S-Sulfenylation, S-Sulfhydration and Sulfinic acid of cysteines. Two missed cleavages of specific trypsin digestion were permitted. Both PSM and protein FDR were set to 1%.

**Analysis of oxidative and total proteome and phosphoproteome.** SILAC quantification of total proteome were log transformed (base 2), median centred, and scaled within each treatment/condition. To characterise the changes in protein abundance in oxidation induced cells compared to control cells, log2 fold changes of 2 and 24 h BCNU or BCNU/AF-treated cells with respect to control cells were estimated using Limma R package[75].

Oxidative profiling data were analysed in a proteome-wide peptide-centric manner in that one or more cysteine oxidation sites from a peptide were quantified simultaneously. Quantifications were first log transformed (base 2), median centred, and scaled within to each treatment/condition. Centred and scaled log quantifications from 24 h BCNU or BCNU/AF treated cells were further normalised by dividing total protein levels from our SILAC proteome quantification data at the same time point (24 h) with the corresponding treatments (BCNU or BCNU/AF) to account for changes in protein abundance. Acute oxidative profiling data (2 h BCNU or BCNU/AF treatments) were excluded from this normalisation procedure, giving that almost no changes were detected on

protein levels under 2 h treatments of either BCNU or BCNU/AF. Next, missing values from oxidative peptides with less than 50% of missingness across samples were imputed using a k-nearest neighbour (k = 10) imputation strategy[76]. Those that have more than 50% missing values were left unchanged.

Phosphosites quantified in at least one sample were log2 transformed, median centred, and scaled within each treatment/condition. Missing values were then imputed for each phosphosite based on its pattern of missingness in the basal and insulin treatments. Specifically, if phosphopeptides were quantified in at least two out of four biological replicates in the insulin-treated samples, and no corresponding phosphopeptide was quantified in any of the four basal samples, a random-tail imputation procedure implemented in the Perseus environment[77] was used to simulate low values in the basal samples alone. This procedure was repeated vice versa for basal compared to insulin samples.

**Visualisation and differential expression analysis.** We employed a probabilistic principal component analysis (PCA) using EM optimisation[78] to generate PCA plots for SILAC proteome, oxidative proteome, and phosphoproteome, respectively. For identifying differentially regulated proteins, oxidative peptides, and phosphosites, normalised and imputed data from SILAC proteome, oxidative proteome, and phosphoproteome were fed into Limma R package for differential regulation analysis. Moderated t-test and empirical Bayesian models were used to fit each dataset and an FDR-adjusted p-value of <0.05 was used as a statistical threshold for identifying significantly regulated proteins, oxidative peptides, and phosphosites. In addition, oxidative peptides regulated in 2 h or 24 h BCNU and BCNU/AF treatments were analysed in combination using a direction-based integrative analysis (direction pathway analysis)[32] to identify concordantly up- and downregulated oxidative peptides and those that are regulated discordantly.

**Pathway analysis and functional annotation.** To perform pathway analysis on the redox proteome and phosphoproteome, we first mapped these data in a protein-centric manner. Specifically, oxidation and phosphorylation levels of a given protein were quantified by the levels of oxidation and phosphorylation of the peptides quantified from the corresponding protein in the redox proteome and phosphoproteome experiments, respectively. For the redox proteome, proteins concordantly up-regulated under the 2 h and 24 h BCNU/AF treatments from direction pathway analysis[32] were tested for enrichment against a Gene Ontology database. For the phosphoproteome, over-representation analysis was performed on phosphoproteins regulated by oxidative stress with 2 h BCNU/AF treatment in the absence of insulin using DAVID[79] (v6.8 beta, released in May/2016, https://david-d.ncifcrf.gov/).

To further annotate function, the kinase, substrate, phosphatase, oxidative proteome profiles were compared with the PhosphoSitePlus database[80] and DEPhOsphorylation database[81]. Specifically, proteins containing significantly altered oxidation site(s) by the treatment of 2 h BCNU/AF were annotated as kinases, phosphatases, substrates of kinases, or others. To generate the circos heatmap plot, we performed hierarchical clustering of proteins with altered oxidation levels upon treatment with 2 h BCNU/AF and overlaid quantification of changes to total protein and phosphorylation levels under various treatments using a gene-centric approach. For kinases with altered oxidation by the 2 h BCNU/AF treatment we visualised phosphorylation profiles of their putative substrates (PhosphoSitePlus) as heatmaps. For the kinases Akt, mTOR, and AMPK, computationally predicted kinase-substrates[40] were additionally utilised for annotation. The UniProt (05/2018) annotations redox-active, interchain and other disulfide forms were used to annotate oxidation sites.

**Structure of Akt1 PH domain and its complex with PIP3.** The crystal structure of Akt1 PH domain was determined at 1.65 Å resolution and is available from the protein data bank (PDB ID: 1UNP). The C60S, C77S and C77F mutations were performed on this structure using the mutator plugin in VMD[82]. The WT and mutant 1UNP structures were used in MD simulations to investigate conformational changes caused by the mutations. To study the effect of the conformational changes on binding of PIP3, we determined the complex structures of Akt1 PH domain and its mutants bound to PIP3 using the methods developed for toxin binding to ion channels[83]. Initial poses for the complexes were obtained using the docking program HADDOCK[84], which were then refined in MD simulations where the complex structures were gradually relaxed as described below.

**MD simulations.** MD simulations were performed using the NAMD program[85] with the CHARMM36 force field[86]. An NPT ensemble with periodic boundary conditions was employed in the MD simulations. The pressure and temperature were maintained at 1 atm and 300 K, respectively, via Langevin coupling with damping coefficients of 5 ps$^{-1}$. Lennard-Jones interactions were switched off smoothly within a distance of 10–13.5 Å, and the particle-mesh Ewald algorithm were applied to calculate the long-range electrostatic interactions. A time step of 2 fs was used to generate the trajectory data in MD simulations.

The simulation systems of Akt1 PH domain for both WT and mutants were prepared using the VMD software[82]. Water box size for all systems were selected so that there were at least three layers of waters between the protein and box boundaries. The simulation box size was $102 \times 89 \times 106$ Å$^3$ for all systems. The

systems were ionized and neutralized with 0.1 M of KCl solution. In MD simulations, the systems were first equilibrated with restrained protein atoms at 300 K to obtain the correct water density with 1 atm pressure coupling. Then the side chain and backbone atoms were simultaneously relaxed in several steps from $k = 5$ kcal/mol/Å$^2$ to 0, simulating the systems for 200 ps at each step. Production data were generated from 100 ns MD simulations for all systems except for C77F, which had to be run for 200 ns due to slower equilibration of the F77 side chain. The trajectory data were analysed using the built-in functions of VMD. The MD simulations for the C60/77S and C77F mutant systems were replicated three times with different initial conditions to ensure that the effect of the mutations were reproducible. As shown in Supplementary Table 2, all three simulations yielded similar results for the E17–R86 distances and loop 1 RMSDs, which were used to identify the conformational changes in loop 1.

Representative images were generated using UCSF Chimera 1.14.

**Soft agar assay**. The soft agar assays for anchorage-independent cell growth were performed as described previously[87]. Briefly, 12-well plates were used and the solid medium consisted of two layers. The bottom layer contained 0.8% noble agar and the top layer contained 0.4% agar suspended with 2000 cells. Here, HEK293 cells overexpressing Akt2-WT, Akt2-W80A mutant or Akt2-C60/77S-W80A mutant were used. DMEM was added twice a week to keep the top layer moisture for the first week and then DMEM with 10 µM MK-2206 was used instead for the next 2 weeks. Cells were stained with 1 mg/mL iodonitrotetrazolium chloride.

**Colony formation assay**. HeLa cells overexpressing Akt2-WT, Akt2-W80A mutant or Akt2-C60/77S-W80A mutant were seeded into 6-well plates (600 cells/well) and fed for 10–12 days in DMEM containing 10% FBS and 2 mM GlutaMAX. Cells were then treated with 10 µM MK-2206 for another 4 days. Colonies were washed with PBS and fixed (10% acetic acid, 10% methanol) for 20 min, and then stained using 0.4% crystal violet in 20% ethanol for 20 min.

**Hoechst assay**. The Hoechst assays for cell proliferation were performed as described previously[88]. Briefly, HeLa cells overexpressing Akt2-WT, Akt2-W80A mutant or Akt2-C60/77S-W80A mutant were seeded into 96-well plates (3,000 cells/well) and fed overnight in DMEM containing 10% FBS and 2 mM GlutaMAX. Cells were then treated with 10 µM MK-2206 for 1 or 2 days. Media was aspirated, and the plate was frozen at −80 °C. Plates were then thawed at room temperature and 100 µL of water was added per well before freezing once more at −80 °C. Plates were thawed, followed by the addition of 100 µL Hoechst-33258 solution, containing 10 µg/mL Hoechst-33258 in TNE buffer (10 mM Tris-HCl, pH 7.4, 2 M NaCl, 1 mM EDTA). The fluorescence was measured at FEx = 360 nm and FEm = 440 nm.

**TIRF experiments**. TIRF experiments were performed as described previously[53]. In brief, 3T3-L1 adipocytes at day 7 were trypsinized, electroporated with the construct of interest and seeded onto Matrigel coated 35 mm µ-dishes (Ibidi). After 24 h, cells were incubated with basal medium (DMEM without FBS) for 2 h. Following this media was replaced with KRP + buffer (KRP, 10 mM glucose and essential amino acids) and dishes were placed onto the stage of a Nikon TiE microscope equipped with an OKOlab microscope enclosure maintained at 37 °C. TIRF was achieved using a Nikon hTIRF module. TagRFP-T was stimulated with a 568 nm laser angled at 71 °C and emission was captured on an Andor 888 emCCD camera after passing through a 610/50 nm filter. Buffer switching was performed using a custom fluidic setup. Cells were treated with insulin, 100 µM BCNU and 1 µM AF, 100 µM H2O2, 50 mU/ml glucose oxidase (GOD), 10 µM diphenylene iodinium (DPI), where added as indicated. Image data were analysed using Fiji[89].

**SDS-PAGE and Western blotting**. Protein concentrations were determined via BCA assay and 10 µg proteins were resolved by SDS-PAGE. The gels were transferred to PVDF membranes. Membranes were blocked in 5% skim milk powder in TBST (0.1% Tween-20 in Tris-buffered saline) for 2 h followed by an overnight incubation at 4 °C with indicated primary antibodies. Membranes were incubated with an appropriate secondary antibody at room temperature for 1 h before signals were detected using a Chemidoc or LICOR imaging system. Immunoblots were quantified by ImageJ software and statistical significance was assessed using Student's t test. All primary antibodies were used at a dilution of 1:1000 unless otherwise stated. Secondary antibodies were used at 1:5000.

**Prdx2/3 assay**. Prdx2/3 assays were performed as described previously[26]. Briefly, cells were washed three times with ice-cold PBS containing 100 mM N-ethylmaleimide (NEM). Cells were scraped in PBS containing 1% (w/v) SDS, protease inhibitors and 100 mM NEM. Samples were centrifuged at 16,000 × g for 20 min at room temperature. Supernatant was collected and protein concentration was determined. Proteins were then resolved by non-reducing SDS-PAGE and immunoblotted using indicated antibodies.

**Protein-lipid overlay assay**. HEK293 cells were transfected with FLAG-Akt2 PH domain using Lipofectamine 2000 and overexpressed FLAG-Akt2 PH domain was purified as described above 48 h post-transfection. The purified FLAG-Akt2 PH domain solution was made into 2 aliquots, one was incubated with 10 mM DTT at room temperature for 20 min, and the other one was incubated with 100 µM H2O2 at room temperature for 20 min.

Protein-lipid overlay assay was performed as described previously[90]. Briefly, a series of lipid dilutions (final amounts 100, 50, 25, 10, 5 and 1 pmol, respectively) were spotted onto nitrocellulose membranes and dried at room temperature for 1 h. after 1 h blocking at room temperature with blocking buffer (50 mM Tris-HCl, pH 7.5, 150 mM NaCl, 0.1% Tween 20, 2 mg/mL fatty acid-free BSA), the membranes were incubated with reduced or oxidized FLAG-Akt2 PH domain in the fresh blocking buffer overnight at 4 °C. Following extensive washing, the membranes were incubated with anti-FLAG antibody (1:1000) at room temperature for 2 h, washed, and incubated with an appropriate secondary antibody at room temperature for another 1 h. Signals were detected and analysed as described above in Western blotting.

**Fly stocks**. The fly stocks used this study were W1118 (v6000, VDRC, Vienna, Austria), dAkt GD1361 (v2902, VDRC, Vienna, Austria), and Tubulin-Gal4/TM3-GFP[91] (gift of Essi Havula, University of Sydney, Australia). The transgenic flies created and crossed for rescue experiments are described below. All experiments were conducted at 25 °C incubators with 65% humidity in 12L:12D cycles and flies were maintained on standard molasses food.

**Genetics**. UAS-hAKT2 WT and UAS-hAKT2 C60/77S were crossed with UAS-dAKT GD1361 to generate UAS-hAKT2 WT; UAS-dAKT GD1361 and UAS-hAKT2 C60/77S; UAS-dAKTGD1361 flies.

For knockdown experiments male UAS-hAKT2 WT; UAS-dAKTGD1361 and UAS-hAKT2 C60/77S; UAS-dAKT GD1361 were crossed to 20 Tubulin-Gal4 females. The full genotypes for these are W1118; UAS-hAKT2-WT/+; UAS-dAKTGD1361/tubulin-GAL4 and W1118; UAS-hAKT2-WT/+; UAS-dAKTGD1361/tubulin-GAL4.

For overexpression experiments UAS-hAKT2 WT and UAS-hAKT2 C60/77S males were crossed directly to Tubulin-Gal4 virgins. The Full genotypes are W1118; UAS-hAKT2 WT; +/tubulin-Gal4 and W1118; UAS-hAKT2 C60/77S; +/tubulin-Gal4. Progeny were reared at standard density under 12L:12D at 25 °C and allowed to mate for 48 h prior to experiments.

**Length and body weight measurements**. Body weight measurements of six males and six females of each genotype were recorded from pre-blanked eppendorf tubes weighed on a scale (Shimadzu ATX224, Tokyo, Japan). Length was measured from micrographs taken using the Zeiss Axiocam ERs 5S mounted on the Zeiss Stemi 305 dissecting microscope (Ziess, Germany) Length measurements were traced using ImageJ/Fiji software (NIH, Bethesda, MD, USA) and all data analysis was performed using Excel and Prism software.

**Glucose treatment for immunoblotting**. Five males and females were starved overnight (16 h) prior to being collected for immunoblotting. Following starvation, batches of flies were administered 10% glucose for 30 and 60 min. Starved and glucose treated flies were homogenized in 2% SDS buffer directly following each treatment.

**Generating UAS-hAKT2 flies**. We use the following primers to generate the UAS-hAKT2 WT and UAS-hAKT2 C77/60S flies: ATCTCGAGATGGACTACAAAGACCATGACG and ATGTTAACCAAGAAAGCTGGGTCTCACTC. The PCR products were digested with XhoI and HpaI and ligated into pJFRC-MUH (pJFRC-MUH was a gift from Gerald Rubin (Addgene plasmid # 26213))[92]. The pJFRC-MUH-hAKT2 WT and pJFRC-MUH-hAKT2 C77/60S constructs were injected into y$^1$ w$^{67c23}$; P{CaryP}attP40 strain embryos (Bestgene, Chino Hills, CA, USA).

**Redox state of Akt Cys60-Cys77 disulfide bond in fibroblasts**. Full-length human Akt1 containing an N-terminal hemagglutinin (HA) tag was expressed in NIH/3T3 fibroblasts using lentivirus. Serum-starved cells ($2.5 × 106$) were incubated with 10 nM human insulin for up to 60 min. At discrete times, whole cell lysates were prepared in phosphate-buffered saline containing 5 mM EDTA, protease inhibitor cocktail (Roche), phosphatase inhibitor (Roche) and 5 mM 2-iodo-N-phenylacetamide (12C-IPA, Cambridge Isotopes) to alkylate the unpaired Cys thiols in Akt1. Akt1 was immunoprecipitated from lysates using an anti-HA antibody (ThermoFisher Scientific, 26183) and protein G dynabeads and resolved on SDS-PAGE. The Akt1 band was excised, disulfide bonds in the protein reduced with dithiothreitol and the new unpaired Cys thiols labelled with a stable carbon-13 isotope of IPA (13C-IPA). The protein was digested with trypsin, cysteine-containing peptides resolved on liquid chromatography and their identity determined by mass spectrometry (Supplementary Fig. 7). The redox state of the cysteines was calculated from the relative ion abundance of 2–4 peptides labelled with 12C-IPA and/or 13C-IPA as we have described[93,94].

**Mechanistic modelling**. The mechanistic Akt model was formulated using ordinary differential equations (ODEs). The model's schematic diagram containing

all model reactions is given in Fig. 7c. The model was implemented in Matlab and model fitting was performed using genetic algorithms as part of Matlab's Optimisation Toolbox. Detailed description of model equations, model fitting procedure and parameter values used for simulations are given in the Supplementary Note 1 and Supplementary Tables 3-5.

**Reporting summary**. Further information on research design is available in the Nature Research Reporting Summary linked to this article.

## Data availability

Data supporting the findings of this manuscript are available from the corresponding author upon reasonable request. A reporting summary for this Article is available as a Supplementary Information file. The source data underlying Figs. 1b–g, 2b–d, 3, 4c–e, 3b, c, 4a–e, g, i, k–n, 5d–l, 6a–i, k, 7a, b, d, e, f, h, i and Supplementary Figs. 1c, d, 2a–g, 3a–d, 5a–i, 6a–g are provided as a Source Data file. All MS data have been deposited to the ProteomeXchange Consortium via the PRIDE partner repository[95] with the dataset identifier PXD011525.

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

## Acknowledgements

We thank Stuart Cordwell, Ben Crossett and Angela Connolly from Mass Spectrometry Core Facility (USYD) for mass spectrometric assistance. The authors acknowledge the facilities and the scientific and technical assistance of Microscopy Australia at the Australian Centre for Microscopy & Microanalysis at the University of Sydney. This work was supported by National Health and Medical Research Council (NHMRC) project grant GNT1120201 (to D.E.J.), Australian Research Council (ARC) Discovery Project DP180103482 (to D.E.J. and J.G.B.) and Australian Research Council (ARC)/Discovery Early Career Researcher Award DE170100759 (to P.Y.). D.E.J. and P.H. are NHMRC Senior Principal Research Fellows. L.K.N. was supported by the Victorian Cancer Agency Mid-Career Research Fellowship (MCRF18026). The contents of the published material are solely the responsibility of the individual authors and do not reflect the views of the NHMRC.

## Author contributions

Z.S., J.G.B. and D.E.J. conceived the project, Z.S., K.F.W. and S.J.H. designed MS experiments. Z.S. prepared samples. Z.S. and B.L.P. performed MS analysis of redox proteome and total proteome. S.J.H. performed MS analysis of phosphoproteome. P.Y., Z.S., J.G.B. and F.V. analysed the data and performed bioinformatics analysis. Z.S. performed biochemical experiments. S.Y. and M.A. performed and analysed the MDS experiments under the supervision of S.K. G.Y., J.S. and Z.S. performed and analysed Western Blots. J.G.B., D.N. and A.K. conducted TIRF experiments and analysed the data. D.F., Z.S. and Q.W. performed fly experiments under the supervision of G.N. J.C. and R.Y. measured the insulin stimulated disulfide in NIH/3T3 cells under the supervision of P.H. S.S. performed the mechanistic modelling under the supervision of L.N. D.E.J. supervised the study. Z.S., P.Y., J.G.B., S.J.H. D.F. and D.E.J. wrote the manuscript with the input of all other authors. All authors approved the final version of the manuscript.

## Competing interests

The authors declare no competing interests.
