## [Peer Review File · Nature Communications]

Reviewers' comments:

Reviewer #1 (Remarks to the Author):

This publication investigates the link between cell signaling mediated by redox modifications of proteins and phosphorylation-based signaling. The authors use adipocytes as their cellular model. They manipulate the redox status of cellular proteins with inhibitors of thiol-based antioxidants systems (glutathione and thioredoxin) which lead to the accumulation of reactive oxygen species (ROS) in the cell. This cellular model has been developed by the authors previously (ref. 26) and in the present study they combine it with mass spectrometry (MS)-based proteomics to quantify the redox proteome as well as the oxidative stress-regulated phosphoproteome.

The authors hypothesize that ROS and phosphorylation could be linked through oxidative stress-induced change in the activity of kinases and/or phosphatases. In particular, the authors focus on the kinases Akt, mTOR and AMPK. They observe that when Akt is oxidized at cysteine C77, it affects phosphorylation of some of its substrates, despite Akt being in its activated (T309 phosphorylated) form. The authors then set out to understand the molecular mechanism by which oxidation of Akt, on cysteines situated in its PH domain, affects its capacity to phosphorylate downstream substrates and they propose an oxidation-dependent mechanism of recruitment of Akt to the plasma membrane.

The focus of this reviewer was on the molecular dynamics simulations and their apport in the mechanistic understanding. Detailed comments are presented below. However, as a general comment, the manuscript is not always very easy to follow and overall clarity could be improved. Also, there was no discussion of whether the observed redox and phospho changes were truly specific or could be simple promiscuous PTMs (see for example Kanshin E. et al., Cell Reports 2015, doi: 10.1016/j.celrep.2015.01.052.)

Detailed comments

The author's viewpoint is that oxidative stress inhibits the ability of Akt to phosphorylate a selected number of substrates (Figure 3), and they observe selective effect of oxidation on substrates for mTOR and AMPK as well (Figure 3). The authors set out to identify the specific oxidative modifications of Akt and identify a disulfide bond C60-C77 in the PH domain, as well as a disulfide bond C297-C311 in the kinase domain of Akt. Both disulfide bonds had been previously identified

(refs. 42-47) and the disulfide bridge C297-C311, situated in the T loop of the kinase, had been associated with kinase inhibition.

The authors then set out to clarify the impact of the C60-C77 disulfide bridge on Akt activity.

As part of their functional investigations, the authors carried out molecular dynamics simulations of the Akt1 PH domain. The goal of the simulations was to understand the effects of Cys oxidation on the structure and interactions of the PH domain with phosphatidylinositol-3-phosphate (PIP3). Calculations were performed introducing Cys to Ser mutations and Cys to F mutations to explore the effects of a pathogenic mutation on the structure of the Akt1 PH domain.

There are several important technical issues with the MD simulations:

- In each case, only one simulation was run, of moderate length (100 or 200 ns). This raises the question of whether the authors get similar results if they run a second simulation starting from different initial conditions, such as the distribution of velocities? The system is not very large, so computational resources should not be an issue. The authors should give more technical details about the simulations (such as the final box dimensions) and perform replicate simulations.
- The authors never state how they determined protonation states of the His residues? Interestingly, His 89 and His 13 are adjacent to one another in the structure. How their protonation states are chosen could influence the structural dynamics of loop 1. Also, one of these His from a neighboring protein within the crystal lattice interacts with PIP3 of the primary asymmetric unit, which may affect the initial structural arrangement. The authors chose to mutate the Cys residues to Ser for subsequent comparison with experimental data on Cys to Ser mutants. However, an important control would be to also run MD using the protonated forms of Cys, as this would better model the non-oxidized system.
- The authors talk about “significant” conformational changes due to mutation from Cys-Ser. How does VMD introduce point mutations? If just by straightforward replacement, could the observed conformational changes be simply due to a steric clash between the mutated residue and the partner Cys because the mutation was built in the same conformation as the original Cys? Did the authors try to locally optimize the sidechain structures of the Ser and partner Cys before starting their simulations?
- Did the authors attempt to take the wild-type protein in apo form and dock the PIP3 to the binding site so see if their simulation can reproduce the conformation changes observed in the crystal structures?
- On page 11, the conclusion that an increase in vibrational energy and its transfer to the ligand disrupts binding and increases structural fluctuations are without any basis. At a minimum, the authors should calculate the RMS Fluctuations on a by-residue basis to show an increase in binding pocket fluctuations, if there is one. Did the authors examine the ligand dynamics? Did the

ligand even remain in the binding pocket during the simulations? I raise this question because the crystal structure suggests that the ligand is stabilized by an interaction with a His in a neighboring protein in the crystal matrix.

- On page 11, the authors mention rotation freedom... rotational freedom of what? The sidechains? The authors could show that through an examination of the trajectories generated.
- The authors write about changes in binding interaction, but they never calculate the binding free energy. There are many simple models based on an MM/GBSA approaches that can provide some quantification.
- Likewise, in the simulation of the C77F mutant (pg.13), the authors, from observation, claim there was a decrease in localized vibrational energy and increased stability of PIP3 binding. Nowhere in the manuscript is there an attempt to quantify changes in vibrational energy and again, no attempt to quantify the binding free energy in the presence of this point mutation.

Overall, the simulations lack rigor and should be subject to a more in-depth analysis. The current MD analysis is too superficial to fully support the authors claim that the disruption of the C60-C77 disulfide bond would impair PIP3 binding.

Finally, at least from this referee's viewpoint, the subsequent redox analysis of the PH domain and analysis of mutants of C60-C77 to S mutants is not clearly presented.

The discussion section on the other hand is clearly written. The authors integrate the ensemble of information in a coherent picture of redox and phosphorylation cross-regulation. However, at least in this referee viewpoint, the data presented may not be entirely sufficient to support the interpretation presented. To sum up, the authors should carefully revise their manuscript and discuss the data more thoroughly to convince the reader of the validity of their proposed interpretation.

Reviewer #2 (Remarks to the Author):

Su et al. performed multi-omic analyses of proteome, redox proteome, and phosphoproteome under oxidative stress, and identified crosstalk between protein oxidation and phosphorylation. Besides being a rich data resource, more importantly, this study revealed that a disulfide bond (C60-C77) localized at the PH domain of Akt plays important roles in its functional regulation. In general, this is a rigorous study with appropriate experimental design, controls, and validations to support the authors' conclusions.

Some important questions concerning this study I detail below.

1. The disulfide bond (C60-C77) was identified by pLink-SS. However, the tandem MS is not of high quality based on Fig. S4c. First, the enzyme trypsin rarely cleavages at K/R residues if a disulfide bond closely follows. In this case, R76-C77 theoretically is not easily to be cut if a disulfide bond presents at C77. More, the mapped peaks are generally very low-abundance. The authors should carefully examine the mass accuracy and E-value for this identification. What protein database (large or small?) was used for pLink-SS? A validation using synthetic peptides should be conducted to confirm this disulfide bond.
2. BCNU is potentially able to react with cysteine directly. Beside the ROS effect caused by BCNU, the quantified proteome and redox proteome could be directly affected by the BCNU modified cysteine. Have the authors examined the modification of BCNU on cysteine?
3. In the redox proteome analysis, is the first step alkylation of free Cys using IAA complete? Are there any Cys residues could be modified by both IAA and iodoTMT? The authors should include data for both IAA and iodoTMT labeled Cys-containing peptides, though IAA-peptides cannot be quantified.
4. The method of redox proteomic experiment was not well described. The authors used 6plex-iodoTMT, however, the "method section" says "4 replicates were performed including 2 forward

labelling assays and 2 reverse labelling assay" (Page 19), which is confusing. The labeling strategy needs to be further clearly explained.

5. Are the values shown in supplementary data1 original intensities of the iodoTMT reporter ions? The variation of this dataset is much higher than what the authors presented in Fig 1C. I understand that we can't compare the absolute intensities in MS2-based quantification for individual proteins. However, the overall (total or average) intensity between different replicates should be consistent. Taken BCNU/AF_24h as an example, the average intensity of BCNU/AF_24h_1 and BCNU/AF_24h_4 are 7359 and 28970, respectively, suggesting that the measurements were poorly repeated. The authors should explain how the original intensities are transformed, normalized, and used to calculate the CVs.

6. The authors highlighted several proteins (e.g. Gsr, Txn2, Prdx5, Txn, Akt, mTOR, AMPK) with changed oxidation level. The quantitative mass spectrometry data, for all detected Cys sites under all conditions, for these proteins should be shown. This reviewer has an impression that different treatments could have varied influence on different sites. For example, Gsr has five oxidation sites as shown in supplementary data 1 and 5. Though C54 and C59 had decreased oxidation level, another Cys site between 321 and 331 increased oxidation with the treatment.

7. The authors found a negative correlation between protein oxidation and protein abundance (Supplementary Fig. 3a). The ROS stress might induce many uncommon/unknown protein modifications, which were not included in the protein database search. As a result, uncommonly modified peptides failed to be identified by database search under oxidative condition, making it look like the protein abundance decreased with oxidation. The authors should check the quantification results using non-Cys-containing peptides to exclude this possibility.

8. In the comparisons between WT and C60/77S mutants in adipocytes, HEK293 cells, Hela cells, and flies, no evidence was shown that the WT Akt indeed had a disulfide bond between C60-C77.

9. Fig 1f. What do the dots with different colors represent?

Reviewer #3 (Remarks to the Author):

In this manuscript the authors use global proteomic methods to study the impact of oxidative stress. They conclude that oxidation of critical cysteines in signaling kinases as well as in the targets of these kinase, modulate signal transduction in a specific and biologically relevant way. The results of studies of specific cysteines in AKT are presented in support of their general conclusion. Overall the data are novel, provocative and the findings will be of interest to a broad audience. However, I am not convinced the authors have demonstrated that the levels of oxidative stress in their experiments is of a magnitude consistent with “physiologic” ROS, thereby undermining their conclusion that the effects observed reflect a physiologic mechanism for tuning signal transduction. My concerns and specific comments for the authors to consider in revising the work are noted below.

1. It is not clear whether the degree of oxidative stress used in this study is with the “physiological” range. The overall significance of the findings depends on this being the case. The behaviors of the cysteine-mutated AKT suggest this is the case but there are other explanations for those findings. Any additional information the authors can provide that address this issue will enhance the impact of the work.
2. It is a bit of a slog going through the initial description of the system (pages 4 to 6). I appreciate the authors were exploring different conditions and optimizing the approach; however, as currently presented it is not easy to follow and the level of detail is overwhelming (and often unnecessary). Anything the authors can do to simplify this narrative will enhance the impact of the conclusions. Apropos of point #1, on page 6 the authors state that “acute oxidative-stress treatments were used so that the oxidative -stress interactions could be examined at more physiologic ROS levels ...”. That being the case, the authors might consider focusing the presentation of the initial studies (pages 4 to 6) on those conditions that are more in line with physiologic ROS levels (e.g., drop the 24 hr treatments?).
3. It would be useful in Figure 3C to present the insulin/basal ratios for the phospho proteomics to give the reader a sense of scale of the impact of oxidative stress relative the effects of insulin. The data on Figure 3C show that oxidative stress induces, at max, a 50% reduction on average of insulin-stimulated phosphorylation of AKT, mTOR and AMPK substrates. I believe this presentation distracts from the main point because what is important is not the global change but differences for specific sites/substrates, a point they try to make in panel b. The data in panel b might be better served if presented in bar graph form so the reader can more acutely know the fold change per site.
4. Figure 5A is a problem. First, they need to present quantification of repeat experiments along with the representative gel. Second, they need to show the effect of oxidative stress on phosphorylation

in cells without insulin stimulation. Does oxidative stress (BCNU/AF) raise the basal level and co-stimulation of insulin further increases the phosphorylation, or does oxidative stress work in synergy with insulin stimulation? I believe no insulin + oxidative stress data are needed to address that question. Without those data one cannot conclude that oxidation of AKT cysteines results in hyperactivation of AKT. Third, the data need to be better discussed/described. The enhanced pT309 phosphorylation induced by oxidative stress does not appear to be reflected in enhanced AKT activity as measured by substrate phosphorylation, although maybe that will be clear when the data are quantified.

Reviewer #4 (Remarks to the Author):

(This review is mainly focusing on the genetic analysis in *Drosophila*).

Su and colleagues have assessed the redox proteome, the phospho-proteome and the total proteome of adipocytes challenged with oxidative stress. The authors tried to correlate the datasets, focusing on a crosstalk between insulin signaling (Akt, mTOR, AMPK) and oxidative stress. The oxidation of Akt was analyzed in more detail (disulfide bonds C60-C77 and C297-C311, S-glutathionylation sites C124 and C311). The authors performed an in-depth analysis of the C60-C77 disulfide bond in Akt with respect to PIP3 binding and Akt activation. Molecular dynamics simulations suggested that the C60-C77 disulfide enhances the binding of Akt's PH domain to PIP3. The functional importance of C60 and C77 for Akt activation was demonstrated in various assays (effects of overexpression of mutant Akt on phosphorylation of Akt and Akt targets as well as on translocation to plasma membrane in adipocytes). Finally, the *in vivo* relevance of C60 and C77 was shown in HEK293 and HeLa cells and in transgenic *Drosophila*. In flies, the wild-type human Akt2 but not the C60/77S mutant hAkt2 was able to replace the endogenous Akt. The authors conclude that they have identified a new regulatory step in the activation of Akt (C60-C77 disulfide bond formation upon oxidative stress enhances PIP3 binding and thus Akt activation).

To correlate the oxidative stress-induced changes in the redox- and the phospho-proteome is certainly a valid strategy, and the newly discovered mode of Akt regulation is exciting. However, there are certain concerns regarding the functional validation.

Why did the authors use the Akt-W80A MK-resistant system to test how C60 and C77 impact Akt activation? MK-2206 interferes with the binding of the PH domain to PIP3 and C60/C77 are thought to regulate the affinity for PIP3. The experiments are further complicated by the endogenous Akt

proteins. The phospho-specific antibodies in Figure 5a are not able to distinguish between endogenous Akt1, endogenous Akt2 and overexpressed Akt2 (wt or mutant). Thus, the results are difficult to interpret. The overexpressed (tagged?) Akt2 should be immunoprecipitated prior to the Akt2 phospho-site analyses.

The soft agar and colony formation assays are also little conclusive because they build upon the Akt-W80A mutant.

The in vivo testing of the C60/77S mutant hAkt2 in *Drosophila* is also not free of complications. First of all, the authors should state that C77 is not conserved in the single fly Akt homolog, casting doubt on the conservation of the proposed mechanism. Whereas the rescue of dAkt-RNAi by the concomitant expression of hAkt2 but not the C60/77 mutant hAkt2 is convincing, the underlying mechanism remains poorly explained. Why is hAkt2-C60/77S constitutively phosphorylated at T309 (but only when dAkt is knocked down)? Why is the wild-type hAkt2 phosphorylated at T309 and S474 even without glucose stimulation (but only in dAkt-RNAi flies, compare Fig. S6c with Fig. 6e)?

Besides these main issues, there are a number of minor problems:

- It should be mentioned that C124 is not present in Akt1.
- To claim that hAkt2 expression in dAkt-RNAi flies “resulted in normal growth”, the authors have to demonstrate that the rescued flies are of the same size as control flies, and that there was no prolongation of the developmental time.
- Instead of the fresh weight, the dry weight of the adult flies should be measured. The fresh weight is very sensitive to the water content. (However, as the observed differences are huge, there is no doubt about the validity of the findings.)
- The VDRC line used for the RNAi of Akt should be listed as UAS-Akt-RNAi (GD1361; v2902) (or UAS-Akt-IR). Otherwise it might be mistaken for an overexpression line.
- What attP landing site was used to generate the transgenic flies?
- The complete genotypes of the experimental flies should be indicated.
- The title of ref. 81 is inserted in the text.

Reviewers' comments:

Reviewer #1 (Remarks to the Author): (**molecular dynamics simulations**)

The focus of this reviewer was on the molecular dynamics simulations and their apport in the mechanistic understanding. Detailed comments are presented below. However, as a general comment, the manuscript is not always very easy to follow and overall clarity could be improved. Also, there was no discussion of whether the observed redox and phospho changes were truly specific or could be simple promiscuous PTMs (see for example Kanshin E. et al., Cell Reports 2015, doi: 10.1016/j.celrep.2015.01.052.)

Response: To address the readability of the paper we have reordered the second half of the paper to improve the flow. We first focus on the PH domain cysteines and their requirement for Akt function. Finally we explore the role of the disulfide and its requirement in the absence and presence of oxidative stress. We have now explored our proposed mechanism with a simple mechanistic model that supports our hypothesis that the PH domain disulfide regulates PIP3 binding and is both present and required under normal physiological conditions.

Concerning the issue of specific versus promiscuous changes in signaling this is obviously an issue of great interest in the field and the paper by Kanshin et al is certainly one paper that has attempted to address this. We recently wrote a detailed review on a related topic in Science Signaling (PMID:30670635) where we described the concept of the “dark phosphoproteome”. That is to say that many reported protein phosphorylation sites are both without function and indeed without a known kinase. Indeed many kinases in the human genome have no reported substrates. Thus, we feel that we have a long way to go before we can answer this question, which is terribly important for all signaling pathways. We have now included a sentence in the discussion to address this issue (second paragraph of discussion).

Detailed comments

There are several important technical issues with the MD simulations:

- In each case, only one simulation was run, of moderate length (100 or 200 ns). This raises the question of whether the authors get similar results if they run a second simulation starting from different initial conditions, such as the distribution of velocities? The system is not very large, so computational resources should not be an issue. The authors should give more technical details about the simulations (such as the final box dimensions) and perform replicate simulations.

Response: We performed replicate simulations (three) for the mutations C60/77S and C77F using different initial conditions. As shown in Table SX2, the replicate simulations yielded consistent results in regard to their effect on the PIP3 binding pocket formed by the loop1 residues, namely, that C77F stabilizes the PIP3 binding pocket while the other mutations destabilize it. The Methods have been amended to reflect these changes and provide further technical detail.

- The authors never state how they determined protonation states of the His residues? Interestingly, His 89 and His 13 are adjacent to one another in the structure. How their protonation states are chosen could influence the structural dynamics of loop 1. Also, one of these His from a neighbouring protein within the crystal lattice interacts with PIP3 of the primary asymmetric unit, which may affect the initial structural arrangement.

Response: As the reviewer points out H89 and HIS13 are adjacent to one another and indeed the protonation state of these residues could influence the structural dynamics. However, protonation of the Histidines is highly unlikely due to the positive charge in this region. As such we took the default protonation states from the CHARMM force field (i.e. unprotonated His). This has now been clarified in the Methods.

-The authors chose to mutate the Cys residues to Ser for subsequent comparison with experimental data on Cys to Ser mutants. However, an important control would be to also run MD using the protonated forms of Cys, as this would better model the non-oxidized system.

Response: We thank the reviewer for the suggestion and performed MD simulations using protonated C60-C77. The simulations yielded similar results to those of C60/77S mutations indicating that breaking the disulfide bond, and not replacing cysteine with serine, is responsible for the conformational changes in loop 1. These new data have been added to the manuscript (see Figure 5i).

- The authors talk about “significant” conformational changes due to mutation from Cys-Ser. How does VMD introduce point mutations? If just by straightforward replacement, could the observed conformational changes be simply due to a steric clash between the mutated residue and the partner Cys because the mutation was built in the same conformation as the original Cys? Did the authors try to locally optimize the sidechain structures of the Ser and partner Cys before starting their simulations?

Response: As detailed in the Methods section, we followed the standard practice of energy minimization followed by gradual relaxation of the protein to optimise the conformations of mutated side chains before starting the unrestrained MD simulations. The fact that the protonated C60-C77 simulations yield similar results to those of the C60/77S mutations support that the conformational changes are due to the breaking of the disulphide bond, and not due to some steric clashes between the mutated side chains.

- Did the authors attempt to take the wild-type protein in apo form and dock the PIP3 to the binding site so see if their simulation can reproduce the conformation changes observed in the crystal structures?

Response: The apo form of Akt 1 (PDB:1UNP) was used for all simulations and we have now docked the head group of PIP3 to Akt and refined the structure in MD simulations. The binding mode obtained is qualitatively very similar to that of the crystal structure (PDB:1UNQ), e.g., all the Akt residues involved in the binding of Ins(1,3,4,5)-Tetrakisphosphate are also involved in the binding of the PIP3 headgroup (Ins(3,4,5)-triphosphate) in the docked complex but the phosphate partners are interchanged. These data have now been added to the manuscript (Fig 5 m. Table S2)

- On page 11, the conclusion that an increase in vibrational energy and its transfer to the ligand disrupts binding and increases structural fluctuations are without any basis. At a minimum, the authors should calculate the RMS Fluctuations on a by-residue basis to show an increase in binding pocket fluctuations, if there is one. Did the authors examine the ligand dynamics? Did the ligand even remain in the binding pocket during the simulations? I raise this question because the crystal structure suggests that the ligand is stabilized by an interaction with a His in a neighboring protein in the crystal matrix.

Response: The residue-specific RMSDs for the loop1 residues 12-26 have now been included (new data: Figure 5i). These data show that the largest changes occur around the E17 residue in the C60/77S mutant and in Akt containing protonated C60-C77, while fluctuations of these residues in the C77F mutant Akt remain similar to that of WT (new data: Figure 6k). Thus, breaking of the C60-C77 disulphide bond is implicated in the conformational change near the tip of loop1, that forms the PIP3 binding pocket. To see the impact of the conformational changes in loop1 on the binding of PIP3, we docked PIP3 to WT Akt and Akt-C60/77S and performed MD simulations of the complex structures. Comparison of the binding modes obtained from docking of PIP3 to WT and mutant Akt shows that PIP3 makes only a few contacts with the mutant Akt compared to 9 contacts with WT Akt (new data: Table S1). In subsequent MD simulations, PIP3 is found to be stably bound to WT Akt while it is observed to unbind from Akt- C60/77S after a short time (new data: Figure 5j-m). Thus, the conformational changes in loop1 prevents PIP3 making proper contacts with the residues of Akt-C60/77S, inhibiting its binding.

- On page 11, the authors mention rotation freedom... rotational freedom of what? The sidechains? The authors could show that through an examination of the trajectories generated^[JB14].

Response: On page 11 rotational freedom was used to refer to the rotational freedom of the sidechains. On reflection this was inappropriate as this was conjecture and the figures referenced were showing the fluctuations in distance between the alpha carbon of C60 and C77. The use of “structural fluctuations” would have been more

appropriate. Further analysis has revealed that the link between the C60/77S mutation and the conformational changes in loop 1 is demonstrated through the breaking of the E17-R86 ionic bond and the subsequent increases in the residue-specific RMSDs of loop1 residues involved in binding of PIP3. We deleted the third paragraph on page 11 (and the associated figures 4f, g, h), which were too general and not focused on PIP3 binding site.

- The authors write about changes in binding interaction, but they never calculate the binding free energy. There are many simple models based on an MM/GBSA approaches that can provide some quantification.

Response: The PIP3 headgroup has a net charge of -5e. The binding free energy calculations using reliable methods such as free energy perturbation are notoriously difficult for such a highly charged ligand. While the MM/GBSA method is feasible, its level of accuracy undermines its utility (see, e.g., Singh N, Warshel A. Absolute binding free energy calculations: on the accuracy of computational scoring of protein-ligand interactions. *Proteins* 78, 1705 (2010)). In any case, the PIP3 headgroup is observed to unbind from Akt-C60/77S in MD simulations, which makes this irrelevant.

- Likewise, in the simulation of the C77F mutant (pg.13), the authors, from observation, claim there was a decrease in localized vibrational energy and increased stability of PIP3 binding. Nowhere in the manuscript is there an attempt to quantify changes in vibrational energy and again, no attempt to quantify the binding free energy in the presence of this point mutation.

Response: The residue-specific RMSDs of loop1 (new data: Figure 6k) show that the C77F mutation has a stabilizing effect on the loop1 residues. In fact, overall the fluctuations are very similar to that of WT. We have also performed MD simulations for the Akt-C77F—PIP3 complex, which showed that PIP3 remained stably bound. The binding mode of the Akt-C77F—PIP3 complex is found to be very similar to that of the Wt Akt—PIP3 complex (new data: Table SX1), with the same number of contacts. So, we expect them to have similar binding free energies.

Overall, the simulations lack rigor and should be subject to a more in-depth analysis. The current MD analysis is too superficial to fully support the authors claim that the disruption of the C60-C77 disulfide bond would impair PIP3 binding.

Finally, at least from this referee's viewpoint, the subsequent redox analysis of the PH domain and analysis of mutants of C60-C77 to S mutants is not clearly presented.

See the very first response on page 1.

The discussion section on the other hand is clearly written. The authors integrate the ensemble of information in a coherent picture of redox and phosphorylation cross-regulation. However, at least in this referee viewpoint, the data presented may not be entirely sufficient to support the interpretation presented. To sum up, the authors should carefully revise their manuscript and discuss the data more thoroughly to convince the reader of the validity of their proposed interpretation.

We hope that the above changes and our revisions to the manuscript have addressed these key issues.

Reviewer #2 (Remarks to the Author): (Mass Spectrometry)

1. The disulfide bond (C60-C77) was identified by pLink-SS. However, the tandem MS is not of high quality based on Fig. S4c. First, the enzyme trypsin rarely cleavages at K/R residues if a disulfide bond closely follows. In this case, R76-C77 theoretically is not easily to be cut if a disulfide bond presents at C77. More, the mapped peaks are generally very low-abundance. The authors should carefully examine the mass accuracy and E-value

for this identification. What protein database (large or small?) was used for pLink-SS? A validation using synthetic peptides should be conducted to confirm this disulfide bond.

Response: Disulfide bonds adjacent to K/R will indeed induce a certain level of steric hindrance to trypsin. However, this does not completely inhibit cleavage but simply reduces efficiency. When we identified disulfide bonds using pLink-SS, 5 missed cleavages for trypsin digestion were allowed. As the samples used for disulfide bond identification were purified by IP, only human Akt2 protein sequence was used for searching. Results were filtered with an E-value cutoff of 0.01 and a mass accuracy cutoff of 10ppm according to Ref #40. Peptides containing C60-C77 and C297-C311 were identified in all three replicates and passed the above were beyond this threshold. We disagree that the presented spectra were of low quality as we obtained excellent sequence coverage and the matched fragments were amongst the most abundant ions in the MS/MS spectrum. Validation with synthetic peptides would of course further improve confidence but the fact we identified these sequences in multiple replicates with high peptide scores and very low overall E-values (range $7.29e-6$ - $1.16e-20$) suggests this is not necessary. The relevant data have been added to Supplementary Figure 4.

2. BCNU is potentially able to react with cysteine directly. Beside the ROS effect caused by BCNU, the quantified proteome and redox proteome could be directly affected by the BCNU modified cysteine. Have the authors examined the modification of BCNU on cysteine?

Response: The reviewer is correct. BCNU can indeed modify proteins. Whilst assessing the amount and specificity of this modification is of great interest, in the current study we used iodo-TMT to label the Cys and we can't quantify BCNU-modified peptides.

3. In the redox proteome analysis, is the first step alkylation of free Cys using IAA complete? Are there any Cys residues could be modified by both IAA and iodoTMT? The authors should include data for both IAA and iodoTMT labeled Cys-containing peptides, though IAA-peptides cannot be quantified.

Response: Cell lysates were treated with 100 mM IAA at room temperature for 40 min. This is a standard method that is generally considered to block the vast majority of free Cys. Cys cannot be modified by both IAA and iodoTMT. We excluded the ID/quantifications of peptides containing multiple Cys residues with ambiguous site localisation. This includes peptides that would be labelled with both IAA and iodoTMT.

4. The method of redox proteomic experiment was not well described. The authors used 6plex-iodoTMT, however, the "method section" says "4 replicates were performed including 2 forward labelling assays and 2 reverse labelling assay" (Page 19), which is confusing. The labeling strategy needs to be further clearly explained.

Response: We have revised the manuscript to clarify the experimental details. The sentence "In redox proteomic study, 4 replicates were performed including 2 forward labelling assays and 2 reverse labelling assays." has been replaced with "4 replicates were performed with 2 forward labelling and 2 reverse labelling strategies. For the forward labelling, tag-126, -127, -128, -129, -130 and -131 were used to label the samples "Ctrl1", "BCNU 2 h", "BCNU 24 h", "Ctrl2", "BCNU/AF 2 h" and "BCNU/AF 24 h", respectively. For the reverse labelling, tag-126, -127, -128, -129, -130 and -131 were used to label "BCNU/AF 24 h", "BCNU/AF 2 h", "Ctrl2", "BCNU 24 h", "BCNU 2 h" and "Ctrl1", respectively.

5. Are the values shown in supplementary data1 original intensities of the iodoTMT reporter ions? The variation of this dataset is much higher than what the authors presented in Fig 1C. I understand that we can't compare the absolute intensities in MS2-based quantification for individual proteins. However, the overall (total or average) intensity between different replicates should be consistent. Taken BCNU/AF_24h as an example, the average intensity of BCNU/AF_24h_1 and BCNU/AF_24h_4 are 7359 and 28970, respectively, suggesting that the measurements were poorly repeated. The authors should explain how the original intensities are transformed, normalized, and used to calculate the CVs.

Response: We provided the raw data in supplementary data1 which are original intensities of the iodoTMT reporter ions (unlogged and unnormalised). Because the intensities may change in each independent experiment, data pre-processing (e.g. transformation and normalisation) are necessary. In this study, we first log-transformed the original intensity values and then normalised each time point to controls and next median-centered and scaled these normalised values across each replicate in each time point. Fig 1d demonstrates the quality of the data and proper pre-processing. CVs are calculated for these pre-processed values. We have now included these data pre-processing steps in the revised manuscript for clarity.

6. The authors highlighted several proteins (e.g. Gsr, Txn2, Prdx5, Txn, Akt, mTOR, AMPK) with changed oxidation level. The quantitative mass spectrometry data, for all detected Cys sites under all conditions, for these proteins should be shown. This reviewer has an impression that different treatments could have varied influence on different sites. For example, Gsr has five oxidation sites as shown in supplementary data 1 and 5. Though C54 and C59 had decreased oxidation level, another Cys site between 321 and 331 increased oxidation with the treatment.

Response: As requested, for the proteins that are highlighted in the manuscript, we have included all Cys site/peptide quantified in the following heatmap to show the oxidation levels under all conditions. This panel now is also included as Supplementary Figure 3d of the revised manuscript.

7. The authors found a negative correlation between protein oxidation and protein abundance (Supplementary Fig. 3a). The ROS stress might induce many uncommon/unknown protein modifications, which were not included in the protein database search. As a result, uncommonly modified peptides failed to be identified by database search under oxidative condition, making it look like the protein abundance decreased with oxidation. The authors should check the quantification results using non-Cys-containing peptides to exclude this possibility.

Response: As suggested, we have now compared the quantification results using Cys containing and non-Cys containing peptides. We found the distribution of the SILAC quantification ratios are highly consistent between these two types of peptides, excluding the possibility of quantification bias due to Cys or non-Cys containing peptides.

8. In the comparisons between WT and C60/77S mutants in adipocytes, HEK293 cells, Hela cells, and flies, no evidence was shown that the WT Akt indeed had a disulfide bond between C60-C77.

Response: In addition to the IP-MS analysis, which identified the disulfide in HEK cells in the presence of BCNU/AF, we have attempted several methodologies to identify the C60-C77 disulphide bond in Akt under native conditions in cells. In particular we have focused our efforts on the click pegylation technique described by van Leeuwen et al ([10.1016/j.freeradbiomed.2017.03.037](https://doi.org/10.1016/j.freeradbiomed.2017.03.037)). Thus far this has not been successful. However, we believe this is not surprising given the highly reducing nature of the cytosol. We propose that this is a very transient modification that likely influences a very small pool of the total Akt in the cell. This is supported by additional experiments in NIH/3T3 cells, where we observe the disulfide exists in about 0.5-1.5% of the total pool. We have now added these details to the manuscript.

9. Fig 1f. What do the dots with different colors represent?

Response: The colors of the points in Fig 1f represent significantly oxidised or reduced peptides based on the direction of their changes in the two comparisons of 2 and 24 h treatments with BCNU/AF. The statistical method used for testing the combined effect of the two comparisons is described in Yang et al. (PMID: 24167158). We have clarified this in the figure legend of the revised manuscript.

Reviewer #3 (Remarks to the Author):

1. It is not clear whether the degree of oxidative stress used in this study is with the “physiological” range. The overall significance of the findings depends on this being the case. The behaviors of the cysteine-mutated AKT

suggest this is the case but there are other explanations for those findings. Any additional information the authors can provide that address this issue will enhance the impact of the work.

Response: The meaning of “physiological” ROS is hard to define for a number of reasons. First, there are many sources of ROS that emerge from different cellular compartments and so the levels of ROS will differ in these different locations. Second, there are numerous methods for measuring ROS yet each method has its limitations and so this limits the ability to define what is a “physiological” versus “unphysiological” level of ROS. Third, ROS is often produced very transiently and so physiological ROS is not just based on its level but also the time of exposure to the ROS.

It is important to clarify that we describe the BCNU/AF treatment as a physiological source of ROS not as a treatment that produces physiological levels of ROS. The BCNU/AF treatment induces relatively severe oxidative stress in a time dependent fashion. These conditions were specifically chosen to ensure that transient disulfide bonds could be readily detected and it is under these conditions that the formation of the disulfide was initially observed. We now add several additional pieces of information that support a physiological role of the disulphide. First and foremost, we show that the disulfide is present in the absence of oxidative stress and is increased with insulin stimulation. Second, a simple mechanistic model of our proposed mechanism is able to faithfully recapitulate our experimental observations. Last, the model was able to accurately predict that loss of insulin stimulated ROS production via the incubation of cells with DPI would impair Akt recruitment in response to insulin. Together these additional pieces of evidence support a physiological role for the PH domain disulphide.

2. It is a bit of a slog going through the initial description of the system (pages 4 to 6). I appreciate the authors were exploring different conditions and optimizing the approach; however, as currently presented it is not easy to follow and the level of detail is overwhelming (and often unnecessary). Anything the authors can do to simplify this narrative will enhance the impact of the conclusions. Apropos of point #1, on page 6 the authors state that “acute oxidative-stress treatments were used so that the oxidative -stress interactions could be examined at more physiologic ROS levels ...”. That being the case, the authors might consider focusing the presentation of the initial studies (pages 4 to 6) on those conditions that are more in line with physiologic ROS levels (e.g., drop the 24 hr treatments?).

Response: We appreciate the reviewer’s comments on readability and have considered the suggestion of removing the 24 h treatment. We feel that whilst the 24 h time point does not feature in the backend of the manuscript, it is still important to provide the full dataset including all time points from a resource perspective. Additionally, the 24 h time point was critical for defining redox sensitive sites and for subsequent pathway analyses. To improve readability we have revised this section for clarity and brevity.

3. It would be useful in Figure 3C to present the insulin/basal ratios for the phospho proteomics to give the reader a sense of scale of the impact of oxidative stress relative the effects of insulin. The data on Figure 3C show that oxidative stress induces, at max, a 50% reduction on average of insulin-stimulated phosphorylation of AKT, mTOR and AMPK substrates. I believe this presentation distracts from the main point because what is important is not the global change but differences for specific sites/substrates, a point they try to make in panel b. The data in panel b might be better served if presented in bar graph form so the reader can more acutely know the fold change per site.

Response: As suggested, we have now included bar plots of insulin/basal ratios of the phosphoproteomics for Akt, mTOR and AMPK substrates in Figure S3C of the revised manuscript. Nonetheless, we would like to note that Figure 3C was specifically crafted to demonstrate the variability in the effect of ROS on kinase activity towards substrates. Given that different substrates have very different fold responses, this cannot be achieved by only displaying the raw insulin/basal ratios. We hope the reviewer agrees that now by presenting these data together it allows a clearer dissection of the effect of ROS on kinase activity.

4. Figure 5A is a problem. First, they need to present quantification of repeat experiments along with the

representative gel. Second, they need to show the effect of oxidative stress on phosphorylation in cells without insulin stimulation. Does oxidative stress (BCNU/AF) raise the basal level and co-stimulation of insulin further increases the phosphorylation, or does oxidative stress work in synergy with insulin stimulation? I believe no insulin + oxidative stress data are needed to address that question. Without those data one cannot conclude that oxidation of AKT cysteines results in hyperactivation of AKT. Third, the data need to be better discussed/described. The enhanced pT309 phosphorylation induced by oxidative stress does not appear to be reflected in enhanced AKT activity as measured by substrate phosphorylation, although maybe that will be clear when the data are quantified.

Response: We now present a simplified version of this experiment, along with quantification from 3 separate replicates. The experiments in Figure 5 were designed to test the physiological relevance of the cysteines and their oxidation status. The BCNU/AF treatment has been removed from this figure as we demonstrate here that loss of the cysteines blocks activation of the kinase.

Oxidative stress (BCNU/AF) (now Supp. figure 5a) does significantly raise the basal level of Akt phosphorylation at T308 but does not increase phosphorylation at 473 or Akt substrates. This has been addressed in the revised manuscript. Importantly, and as the reviewer points out, the enhanced phosphorylation of T308 is not reflected in increased substrate phosphorylation. This can be seen clearly in figure 3B, where we observe hyperphosphorylation of T309 (Akt2) alongside, largely impaired substrate phosphorylation. We believe this is due to the formation of the T-loop disulfide Cys297-Cys311 identified with the targeted MS (supplementary Figure 4D). The newly added mechanistic model suggests that BCNU/AF and insulin do work synergistically (Supplementary figure 5c), with respect to recruitment of Akt to the PM and phosphorylation at T309.

Reviewer #4 (Remarks to the Author):

Why did the authors use the Akt-W80A MK-resistant system to test how C60 and C77 impact Akt activation? MK-2206 interferes with the binding of the PH domain to PIP3 and C60/C77 are thought to regulate the affinity for PIP3.

Response: To test the Akt mutants we wanted to use a system that is devoid of endogenous Akt as this would confound the interpretation of Akt activity. As the reviewer points out, MK-2206 blocks Akt by locking the PH domain to the kinase domain. Tryptophan 80 (W80) is required for MK-2206 binding and as such the W80A mutation renders Akt insensitive to MK-2206 inhibition. This allows overexpressed Akt containing the W80A mutation to be assessed in the absence of activity of the endogenous Akt isoforms. This is a very clean and powerful system. It is important to note that MK-2206 cannot bind to either the kinase or PH domain alone.

The experiments are further complicated by the endogenous Akt proteins. The phospho-specific antibodies in Figure 5a are not able to distinguish between endogenous Akt1, endogenous Akt2 and overexpressed Akt2 (wt or mutant). Thus, the results are difficult to interpret. The overexpressed (tagged?) Akt2 should be immunoprecipitated prior to the Akt2 phospho-site analyses. The soft agar and colony formation assays are also little conclusive because they build upon the Akt-W80A mutant.

Response: As explained above, the W80A system enables interrogation in the absence of endogenous activity. As demonstrated with WT Akt overexpression the addition of MK-2206 completely blocks the activation and phosphorylation of endogenous Akt. Endogenous Akt is not recruited or phosphorylated in the presence of MK-2206. As such the phospho antibodies are only detecting the overexpressed Akt. Hence, we feel that this obviates the need for immunoprecipitation.

The in vivo testing of the C60/77S mutant hAkt2 in Drosophila is also not free of complications. First of all, the authors should state that C77 is not conserved in the single fly Akt homolog, casting doubt on the conservation of the proposed mechanism.

Response: The reviewer correctly points out that flies are missing C77 (but not C60) and we have now pointed this out in the revised manuscript. This is common in PH domains that bind PIP3 as a whole with the majority having C60, several with C77 but none with both. The second cysteine appeared in Akt prior to the triplication event that occurred before the evolution of higher order vertebrates and mammals (REF) and they have been maintained ever since. This suggests that the CSD in the PH domain is an Akt specific modification that plays an important functional role in higher organisms. We have been unable to find this particular disulfide motif in other PH domains at a structural level, suggesting this is an Akt specific modification that is specific to higher organisms.

The currently accepted mechanism for the evolution of reversible disulfides, is that one cysteine that is redox active and confers functional advantage appears first. This is followed by later acquisition of a second cysteine that allows for disulfide formation and offers further advantage. These points are now addressed in the discussion.

Whereas the rescue of dAkt-RNAi by the concomitant expression of hAkt2 but not the C60/77 mutant hAkt2 is convincing, the underlying mechanism remains poorly explained. Why is hAkt2-C60/77S constitutively phosphorylated at T309 (but only when dAkt is knocked down)? Why is the wild-type hAkt2 phosphorylated at T309 and S474 even without glucose stimulation (but only in dAkt-RNAi flies, compare Fig. S6c with Fig. 6e)?

Response: hAkt2 is not dAkt. The proteins share about 54% sequence homology in their overlapping regions and dAkt has an additional 110 amino acids. As such functional differences between the proteins can be expected. The S505 described in the paper is the homologue of S474 in hAkt2. The homologue of T309 is T342 in dAkt2. We have been unable to source an antibody to this latter site and its regulation is not well described in the literature. The major point we are making here is that the double mutant cannot be activated classically in the overexpressed flies and in the knockout flies as shown by the complete lack of 474 phosphorylation. The refeeding response is however still present for the WT hAkt2. This is consistent with a loss of PM recruitment. The T309 hyperphosphorylation observed particularly with the mutant is likely a compensatory response. T309 is essential for activation of Akt, and Akt activation is essential for survival, without some level of Akt activity the flies would not be viable. It is clear however that despite the hyperphosphorylation of this site that there is a strong functional consequence of loss of the cysteines.

Besides these main issues, there are a number of minor problems:

- It should be mentioned that C124 is not present in Akt1.

Response: We have clarified that cysteine modifications that we identified including the glutathionylation of C124 were found in Akt2.

- To claim that hAkt2 expression in dAkt-RNAi flies “resulted in normal growth”, the authors have to demonstrate that the rescued flies are of the same size as control flies, and that there was no prolongation of the developmental time.

Response: We have changed the wording as the phenotype is best described as a size defect, that may or may not be linked to growth.

- Instead of the fresh weight, the dry weight of the adult flies should be measured. The fresh weight is very sensitive to the water content. (However, as the observed differences are huge, there is no doubt about the validity of the findings.)

Response: As the reviewer rightly points out the differences in size are large and we measured the length and wing size, which are all significantly different. We did not measure dry weight.

- The VDRC line used for the RNAi of Akt should be listed as UAS-Akt-RNAi (GD1361; v2902) (or UAS-Akt-IR). Otherwise it might be mistaken for an overexpression line.

Response: Thank you for this observation. We have subsequently made the suggested changes to the text.

- What attP landing site was used to generate the transgenic flies?

Response: The landing site was on the appt40 site on the 2nd chromosome. This has been added to the methods text.

- The complete genotypes of the experimental flies should be indicated.

Response: We appreciate that the description of the genotypes may have been confusing and we have now added the full genotypes to the methods text.

- The title of ref. 81 is inserted in the text.

Response: This has now been corrected in the text

Reviewers' comments:

Reviewer #1 (Remarks to the Author):

This reviewer focused on the molecular dynamics. The authors have nicely clarified the points raised in the original review. The ensemble of computational and experimental results presents an original insight on the interplay between redox and phosphorylation signalling that is of general interest for understanding crosstalk between signalling pathways.

I only have a few minor points that the authors could clarify.

Minor comments:

Figure 5 and Table S2: The authors performed replicate simulations for the mutants C77F and C60/77S and show that distances of interest are reproducible between the simulations. For sake of completeness, they could have performed replicate simulations of the WT system as well.

Pg. 13 line 357 & 360: The authors sometimes uses notation Table SX1 and sometimes Table S1.

Pg. 13 lines 360-361 There is a detailed list of contacts between the WT Akt and PIP3 for the WT and mutant C77F (Table S1), but only the graphical representation of 3 distances for the mutant C60/C77S (shown in Figure 5 j,k,l). Providing numerical data for all systems in Table S1 would be more complete. The labelling of the atoms in Table S1 (for example, R23 (N1), R23 (NE)) could be made clearer. Likewise the equivalent of Figures 5 d-h and j-l could be provided as Supplementary material for the reduced form of the PH domain as well as for the C77F mutant.

The RMSD per residues (Figure 5 i and 6 k) should be described more clearly: are these RMSD between two structures (initial and final?) or an average RMSD per residue over the course of the trajectory? In the latter case, could error bars be provided?

Could the author comment on why the initial docked structures (Figures 5m, at t=0 ns) are so different between the WT and C70/C77S mutant. They state (pg. 13 line 361) that only 3 contacts anchor PIP3 in the mutant vs. 9 in the WT. However, given that at t=0 ns the PIP3 binding pocket does not yet seem to have opened much, a word of explanation on the origin of the difference in initial docked structures would be useful.

Reviewer #2 (Remarks to the Author):

The authors have addressed most of the points raised by the reviewers. However, some questions that were not fully answered should be addresses before the acceptance.

1. Former remark 2: Since the BCNU can modify proteins directly, more data and discussions need to be provided. In the current manuscript, the authors only explain how BCNU increases cysteine oxidation by accumulating ROS through the inhibition of glutathione reductase, the BCNU direct modification was completely ignored. Is the reversible cysteine modification affected by BCNU irreversible modification? More, though BCNU-modified peptides can't be quantified, the identification of BCNU-modified peptides should be shown to roughly estimate to what extent BCNU can affect the cysteine in a direct manner. The real scenario may be complicated, e.g. a decreased iodoTMT modification may be associated with an altered BCNU direct modification.

2. Technical question: May I know the reason why the authors didn't use iodoTMT antibody to pull-down the iodoTMT-modified peptides to deepen the analysis? The author should show the numbers of PSMs that were derived from iodoTMT-modified and unmodified peptides, respectively.

3. As I commented previously, the identification of the disulfide bond C60-C77 of Akt lacks validation, especially in those biologically relevant samples. The author explained that the disulfide bond between C60-C77 of Akt only accounts for a very small fraction of the total Akt population, thus can't be easily captured. Why does this small group of oxidized Akt have a substantial consequence on overall Akt function? Why can't so much unmodified Akt compensate modified Akt's function?

Reviewer #3 (Remarks to the Author):

In the revised manuscript the authors have sufficiently addressed the prior criticisms and by doing so have improved the clarity and impact of the study. I have no further concerns and I support publication of the work.

Reviewers' comments:

Reviewer #1 (Remarks to the Author):

This reviewer focused on the molecular dynamics. The authors have nicely clarified the points raised in the original review. The ensemble of computational and experimental results presents an original insight on the interplay between redox and phosphorylation signalling that is of general interest for understanding crosstalk between signalling pathways.

I only have a few minor points that the authors could clarify.

Minor comments:

Figure 5 and Table S2: The authors performed replicate simulations for the mutants C77F and C60/77S and show that distances of interest are reproducible between the simulations. For sake of completeness, they could have performed replicate simulations of the WT system as well.

Response: WT replicate data has now been added to table S2

Pg. 13 line 357 & 360: The authors sometimes uses notation Table SX1 and sometimes Table S1.

Response: We thank the reviewer for identifying this. This has been corrected in the text.

Pg. 13 lines 360-361 There is a detailed list of contacts between the WT Akt and PIP3 for the WT and mutant C77F (Table S1), but only the graphical representation of 3 distances for the mutant C60/C77S (shown in Figure 5 j,k,l). Providing numerical data for all systems in Table S1 would be more complete. The labelling of the atoms in Table S1 (for example, R23 (N1), R23 (NE)) could be made clearer. Likewise the equivalent of Figures 5 d-h and j-l could be provided as Supplementary material for the reduced form of the PH domain as well as for the C77F mutant.

Response: PIP3 unbinds from Akt[C60/77S] quickly during MD simulations and the contact distances increase with time. The average distances are not well-defined quantities in this case, and that is why they were not included in Table S1. In contrast, PIP3 binds stably to both Akt[WT] and Akt[C77F] so one can determine the average contact distances from MD simulations. The side chain atoms in the table were labelled following the CHARMM force field (mentioned in the table caption now).

The equivalent of Figures 5 d-h for the reduced Akt have now been included as requested (Supplementary Figure 5a-e). The equivalent of 5e for C77F is present already as figure 6h and the remainder are now included as Supplementary Figure 5 f-i.

Because PIP3 binds to Akt[C77F], the equivalent figures to 5j-l are very similar to the WT and do not add anything to the detail already provided. We have not docked PIP3 to the reduced Akt.

The RMSD per residues (Figure 5 i and 6 k) should be described more clearly: are these RMSD between two structures (initial and final?) or an average RMSD per residue over the course of the trajectory? In the latter case, could error bars be provided?

Response: The residue-specific RMSDs were averages per residue over the course of the trajectory obtained from the MD simulations using the crystal structure as a reference. We have refrained from

showing the error bars in the figures for the sake of clarity but they range from 0.2 to 0.5 Å. This information has been included in the figure captions.

Could the author comment on why the initial docked structures (Figures 5m, at t=0 ns) are so different between the WT and C70/C77S mutant. They state (pg. 13 line 361) that only 3 contacts anchor PIP3 in the mutant vs. 9 in the WT. However, given that at t=0 ns the PIP3 binding pocket does not yet seem to have opened much, a word of explanation on the origin of the difference in initial docked structures would be useful.

Response: The docking was performed as follows

1. Perform mutation
2. Equilibrate the systems in MD, making sure that conformational changes are adequately taken into account. Normally there will be one dominant conformation as was the case here. *(This is t=0 and as such the differences reflect the dominant conformations of WT vs mutant.)*
3. Dock the ligand to this conformation.
4. Perform MD simulations to check the stability of the ligand.

In this case, PIP3 was docked to the well-equilibrated structures. The data here support our prediction that the movement of loop1 and disruption of the E17-R86 ionic bond that we observe in C60/C70S mutant impairs PIP3 binding.

Reviewer #2 (Remarks to the Author):

The authors have addressed most of the points raised by the reviewers. However, some questions that were not fully answered should be addressed before the acceptance.

1. Former remark 2: Since the BCNU can modify proteins directly, more data and discussions need to be provided. In the current manuscript, the authors only explain how BCNU increases cysteine oxidation by accumulating ROS through the inhibition of glutathione reductase, the BCNU direct modification was completely ignored. Is the reversible cysteine modification affected by BCNU irreversible modification? More, though BCNU-modified peptides can't be quantified, the identification of BCNU-modified peptides should be shown to roughly estimate to what extent BCNU can affect the cysteine in a direct manner. The real scenario may be complicated, e.g. a decreased iodoTMT modification may be associated with an altered BCNU direct modification.

The identification of the BCNU modified peptides is not trivial, as it is not clear from the literature what the modifications are. BCNU rapidly decomposes into 2-Chloroethyl diazohydroxide (can react directly with nucleophiles to form 2-chloro-ethylated products) and 2-chloroethyl isocyanate (CEIC-carbamoylation of amino groups). Further decomposition of these along with their adducts also occurs, making mass determination problematic. The inhibition of Gsr is reported to result from the reversible carbamoylation of Cys58 (Cys54 in our data) by CEIC (Eur. J. Biochem. 171, 193-198 (1988)) (Biochem Pharmacol. 1994 Aug 3;48(3):587-94). It has also been suggested that this site could be alkylated which would, as the reviewer implies, be irreversible.

Nevertheless, to answer the reviewer's question, we have re-searched our MS data with several variable modifications. We focused on analysing only the total proteomics data since Cys-containing BCNU adduct modified peptides will not be labelled with iodoTMT. Based on the

above papers we included the following modifications to cysteine: i) (N-acetyl-S-[N-(2-chloroethyl)carbamoylation (ACCC: C[3]H[4]N[1]O[1]Cl[1]), ii) carbamylation (H[1]N[1]O[1]C[1]), and iii) 2-hydroxyethylation (CCOH: H[4]O[1]C[2]). All peptide spectral matches were filtered to 1% FDR using the target decoy strategy in Andromeda/MaxQuant. To further add confidence in assignments, we required modified peptides to be identified by at least 2 MS/MS spectra. These spectra were then manually annotated to ensure >50% of the most abundant fragment ions were assigned to b- or y-type ions within a mass accuracy of 20 ppm. This stringent analysis resulted in identification of only 2-hydroxyethylated cysteines on a total of seven peptides. An annotated MS/MS spectra of the 2-hydroxyethylation of LGALS1 that was identified in all biological replicates is shown below with quantification shown as

an insert. A time-dependent increase in abundance of this peptide was observed following BCNU treatment. Other modified peptides show similar increases but were not quantified in >2 replicates. It is important to note that the other modifications may indeed be induced by BCNU but our ability to detect these are potentially limited. For example, chlorine-modified peptides will produce an isotopic distribution that deviates from theoretical average in isotopic patterns. Our acquisition utilised these theoretical isotopic patterns to trigger ions for MS/MS which is commonly employed to maximise peptide sequencing. Therefore, ions that deviate from theoretical distributions of 'natural' peptides may not be triggered for MS/MS. This is often a default setting in mass spectrometers established for deep proteomic analysis and its removal significantly reduces peptide identification rates.

Although we did not identify modification of a known target of BCNU (Gsr Cys54), we have taken the response profile of this cysteine (Gsr Cys54/59) and searched for cysteines showing similar responses by defining a dissimilarity score based on the Euclidean distance of the log2 fold changes across all treatments. We found that the similarity scores of other cysteines to Gsr Cys54/59 is roughly normally distributed. We therefore fitted a normal distribution and selected the sites that resemble Gsr Cys54/59 using a p-value cutoff of 0.05. This identified 367 cysteines out of a total of 13,199 (2.8%). These sites such as Nit1 Cys166 (the closest) could be considered candidate sites of

BCNU action but will require further validation, that is beyond the scope of this work. None of these sites were detected in our above search for BCNU adducts.

It should be noted that, treatment of cells with BCNU alone, had a modest effect (at best) at 2 h and was reversed by 24h. Only when combined with Auranofin did we observe large changes in cysteine oxidation, and this was foundational to our entire study. Taking all of this together, we conclude that the effect of BCNUylation is unlikely to have had a significant impact on the majority of reversibly oxidised cysteines.

We have now added the following statement to the manuscript “ It has been reported that BCNU can form direct adducts with cysteine residues in Gsr and potentially other proteins [Eur. J. Biochem. 171, 193-198 (1988)]. We performed an extensive search of our MS data for these modifications and were unable to find any changes likely to impact our quantification of oxidised cysteines.”

2. Technical question: May I know the reason why the authors didn't use iodoTMT antibody to pull-down the iodoTMT-modified peptides to deepen the analysis? The author should show the numbers of PSMs that were derived from iodoTMT-modified and unmodified peptides, respectively.

Response: During pilot studies we evaluated the use of the iodoTMT antibody to enrich for TMT-labelled peptides. In our hands, this did not deepen the analysis beyond which we were able to achieve in the absence of enrichment, because losses associated with the IP were substantial. This same finding is confirmed by the vendor (Thermo), who on their webpage present data to show that without antibody-based enrichment, 4,916 iodoTMT-modified peptides could be identified. However, following enrichment with the antibody the number of modified peptides was almost 3-fold smaller (1,709) (source: <https://www.thermofisher.com/au/en/home/life-science/protein-biology/protein-biology-learning-center/protein-biology-resource-library/protein-biology-application-notes/specific-labeling-enrichment-quantitation-on-s-nitrosylated-peptides-using-iodotmt-reagents.html>). This is in-line with our experience. Importantly, the use of an antibody to enrich iodoTMT-modified peptides may also introduce unknown biases, as any sequence preferences of the antibody are unclear. We are therefore satisfied with the unbiased approach we have taken.

As requested we have now included the number of unmodified and iodoTMT-modified PSMs “This led to the identification of 281,345 unmodified and 66,002 iodoTMT-modified PSMs.” in the method section in “Sample preparation, mass spectrometry analysis and data processing for redox proteome” of the revised manuscript.

3. As I commented previously, the identification of the disulfide bond C60-C77 of Akt lacks validation, especially in those biologically relevant samples. The author explained that the disulfide bond between C60-C77 of Akt only accounts for a very small fraction of the total Akt population, thus can't be easily captured. Why does this small group of oxidized Akt have a substantial consequence on overall Akt function? Why can't so much unmodified Akt compensate modified Akt's function?

Response:

“The identification of the disulfide bond C60-C77 of Akt lacks validation, especially in those biologically relevant samples”

We believe there is good evidence to support the existence of this disulphide bond in Akt as follows:

(1) Both Akt disulfides discussed in the paper have been identified in crystal structures (as noted in the manuscript). The C60-C77 disulfide is a cross strand disulphide that meets the criteria defined for a redox active disulfide.

(2) In our primary screen we identified that oxidation of both C60 and C77 increased with BCNU/AF treatment (Fig 3, Supplementary Fig. 3c) as did C311 and C124. IP-MS of Akt then identified the PH domain disulfide (C60-C77), the t-loop disulfide C297-C311 along with glutathionylation of C124 and C311 in BCNU/AF treated cells, confirming its presence under the BCNU/AF condition (Supplementary Fig. 4).

(3) Mutation of either C60 or C70 was sufficient to impair both Akt recruitment and activation. Nevertheless, we do acknowledge that there is still the possibility that modification of either c60 and or c70, independently of disulfide bond formation could still play a crucial role in Akt activation. To acknowledge this point we have now inserted the following statement in the revised manuscript.

As noted in the manuscript “Our data suggest that C77 may play a more central role in Akt activation than C60. One possibility is that C77 oxidation alone, without formation of a disulfide bond with C60, can regulate Akt activation to some extent.” This is based on the C77F mutation and the fact that the C60S mutation maintains about 50% of WT activity.

“The author explained that the disulfide bond between C60-C77 of Akt only accounts for a very small fraction of the total Akt population, thus can't be easily captured. Why does this small group of oxidized Akt have a substantial consequence on overall Akt function?” and “Why can't so much unmodified Akt compensate modified Akt's function?”

We have previously shown that there is considerable sparseness in the activation of Akt (PMID: 22207758) and our TIRF data shows that at physiological insulin very little Akt is actually recruited to the membrane (doi: 10.1242/jcs.205369). Our data support the hypothesis that the disulphide bond is involved in Akt recruitment (Fig 5, Fig 6a-c, Fig 7. a-f.) and maybe dispensable thereafter (once it is released back into the reducing environment of the cytosol). Thus, it makes perfect sense that the stoichiometry of this reaction is very low. As such, this small pool of oxidised Akt is likely to have a tremendous impact on Akt activation since this pool has an increased affinity for PIP3 as shown by Fig 6a,b.

Reviewer #3 (Remarks to the Author):

In the revised manuscript the authors have sufficiently addressed the prior criticisms and by doing so have improved the clarity and impact of the study. I have no further concerns and I support publication of the work.

We thank the reviewer for their comments, which we believe have helped improve the manuscript.